# A Review on Different State of Battery Charge Estimation Techniques and Management Systems for EV Applications

**Girijaprasanna T and Dhanamjayulu C \***

School of Electrical Engineering, Vellore Institute of Technology, Vellore 632014, India;
girijaprasanna.t2019@vitstudent.ac.in
\* Correspondence: dhanamjayulu.c@vit.ac.in

**Abstract:** Electric vehicles (EVs) have acquired significant popularity in recent decades due to their performance and efficiency. EVs are already largely acknowledged as the most promising solutions to global environmental challenges and $CO_2$ emissions. Li-ion batteries are most frequently employed in EVs due to their various benefits. An effective Battery Management System (BMS) is essential to improve the battery performance, including charging–discharging control, precise monitoring, heat management, battery safety, and protection, and also an accurate estimation of the State of Charge (SOC). The SOC is required to provide the driver with a precise indication of the remaining range. At present, different types of estimation algorithms are available, but they still have several challenges due to their performance degradation, complex electrochemical reactions, and inaccuracy. The estimating techniques, average error, advantages, and disadvantages were examined methodically and independently for this paper. The article presents advanced SOC estimating techniques, such as LSTM, GRU, and CNN-LSMT, and hybrid techniques to estimate the average error of the SOC. A detailed comparison is presented with merits and demerits, which helped the researchers in the implementation of EV applications. This research also identified several factors, challenges, and potential recommendations for an enhanced BMS and efficient estimating approaches for future sustainable EV applications.

**Keywords:** electric vehicles; battery management system; Li-ion batteries; algorithms; SOC estimation of battery; accuracy

## 1. Introduction

Nowadays, green environments and environmental hazards are the most significant concerns of researchers [1,2] The world is moving towards severe consequences such as GHG (Green House Gas) emission and global warming caused by the wide use of petrol and diesel in vehicles' operation, which produces lots of carbon dioxide every year [3–5]. The EVs are the most suitable solution to reduce this carbon emission [6–11]. The development of EVs generates massive employment in different sectors of EVs, such as battery manufacturing, powertrain modeling, highly efficient motor designing, etc. Batteries have been widely used at small and medium scales in electricity storage technologies due to their relatively high energy density, low noise levels, and low maintenance [12–15]. Li-ion and Ni-MH batteries are generally utilized in a spread of EV applications. Li-ion plays a vital role because of more advantages such as long life, high efficiency, and energy density, as shown in Table 1 [16–19]. The accuracy of SOC varies depending upon the type of lithium-ion battery, which is heavily impacted by the positive and negative electrode materials.

Commonly used Li-ion batteries for EV applications are lithium cobalt oxide (LCO), lithium titanium oxide (LTO), lithium nickel oxide (LNO), lithium iron phosphate (LFP), lithium manganese oxide (LMO), lithium nickel cobalt aluminum oxide (NCA), and lithium nickel manganese cobalt oxide (NMC). Table 2 depicts a performance comparison of various types of Li-ion batteries [15].

**Table 1.** Comparison of different energy storage devices [16–19].

| Storage Devices | Nominal Voltage (V) | η (%) | Energy Density (Wh/L) | Life Cycle (hrs) | Depth of Discharge (%) | Cost Estimation (USD/kWh) |
|---|---|---|---|---|---|---|
| Lead Acid | 2.0 | 85 | 50–80 | 1500 | 50 | 105–475 |
| NaNiC1 | - | 84 | 160–275 | 3000 | 100 | 315–488 |
| ZBFB | 1.8 | 70 | 55–65 | 10,000 | 100 | 525–1680 |
| Li-ion | 4.3 | 96 | 200–400 | 10,000 | 95 | 200–1260 |

**Table 2.** Comparison of different Li-ion batteries and their characteristics [15].

| Battery Name | Nominal Voltage (V) | Specific Energy (Wh/kg) | Charge (c) | Discharge (c) | Lifespan (hrs) |
|---|---|---|---|---|---|
| LCO | 3.7~3.9 | 150~200 | 0.7~1 | 1 | 500~1000 |
| LNO | 3.6~3.7 | 150~200 | 0.7~1 | 1 | >300 |
| LMO | 3.7~4.0 | 100~150 | 0.7~1 | 1 | 300~700 |
| NMC | 3.8~4.0 | 150~220 | 0.7~1 | 1 | 1000~2000 |
| LFP | 3.2~3.3 | 90~130 | 1 | 1 | 1000~2000 |
| NCA | 3.6~3.65 | 200~260 | 0.7 | 1 | 500 |
| LTO | 2.3~2.5 | 70~85 | 1 | 10 | 3000~7000 |

An effective BMS can work reliably and safely. It is also essential for updating data, controlling the voltage equalizing of a battery, and sensing faults that are substantial influences for attaining a better precision of SOC. The SOC in a BMS is taken into account together with severe and significant issues, which have been investigated in current years. With a gasoline-powered automobile, the SOC of the battery organizes the fuel indicator's similar action, which specifies the quantity of remaining energy in the battery. An accurate estimation of battery states not only provides information about the current and remaining performance of the battery but also ensures the EV's reliable and safe operation. However, estimating battery SOC is one of the most difficult challenges for the successful operation of EVs. Battery SOC cannot be directly observed due to nonlinear, time-varying characteristics and electrochemical reactions [20].

Furthermore, the battery's performance is heavily influenced by temperature variation, aging, and charged–discharge cycles, making estimating an accurate SOC challenging [21]. Very little literature provides a detailed explanation of all methods for SOC estimation for EVs [22–26]. The battery-accurate SOC estimation problem has not been efficiently solved [27,28]. References [29–32] provided a detailed SOC estimation in terms of overall research progress, future development trends, and the source of SOC estimation. However, there is no systematic explanation of the SOC calculation process and algorithm selection and how to deal with uncertain environmental conditions and battery pack grouping in EVs. The literature has illustrated some standard methods for estimating SOC; however, each technique has gaps in terms of accuracy and data availability. Furthermore, complex calculations and high computation costs are two concerns that make the estimation process difficult.

As a result, academics, researchers, and scientists have conducted extensive research to improve the accuracy of battery SOC. Nonetheless, the issues with estimating an accurate SOC have not been resolved. Furthermore, the challenges in estimating the SOC have not been identified. Figure 1 shows the number of research articles on Li-ion battery SOC estimation that have been published, which describes the growing interest in Li-ion battery SOC estimations in recent years. These published research articles were discovered using the Web of Science database and other journals also. From 2006 to 2022, the search criterion was "state of charge", followed by "Li-ion battery". Table 3 summarizes the recently reported studies covered and also presents the review articles' covered aspects.

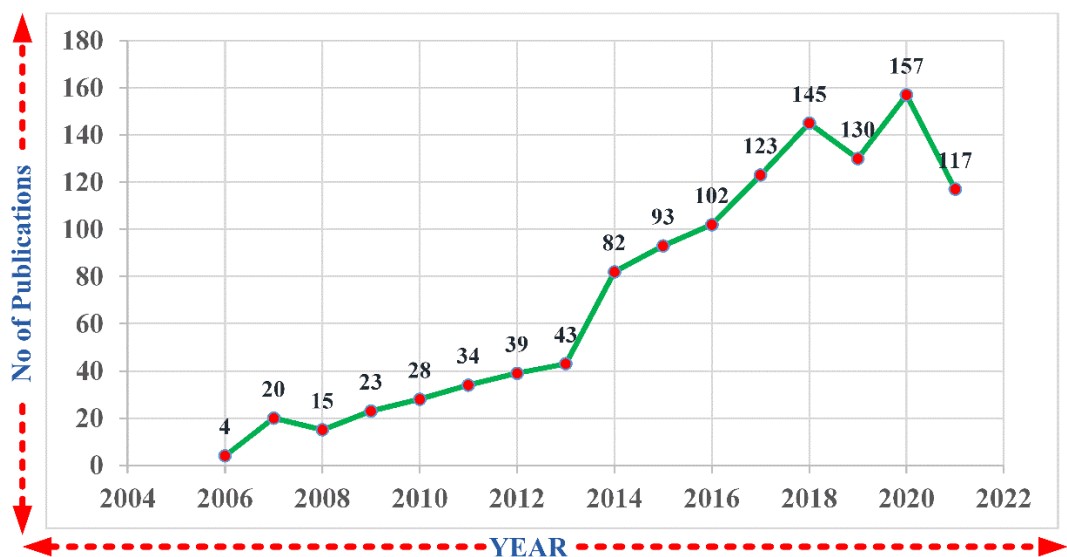

**Figure 1.** The number of research articles on Li-ion battery SOC estimation per year.

**Table 3.** Summary of recently reported studies covered and review articles covered.

| Concentrated Parameters | Recently Reported Studies Covered | | | Present Review Article Covered |
|---|---|---|---|---|
| | [10] | [11] | [12] | |
| Conventional Algorithms | √ | √ | √ | √ |
| Adaptive Filter | √ | √ | √ | √ |
| Learning Algorithms | √ | √ | √ | √ |
| Advanced Techniques | X | x | √ | √ |
| Hybrid | √ | x | √ | √ |
| Advantages | X | √ | x | √ |
| Disadvantages | X | √ | x | √ |
| Applications | X | x | √ | √ |
| Average Error | X | x | x | √ |
| Factors, Challenges, and Recommendations | X | √ | x | √ |
| Future Scope | X | √ | x | √ |

Note: √ = yes and x = no.

As a result, this research paper fills the gap by investigating various existing methodologies and addressing the key issues and challenges associated with SOC estimation. This study will benefit automobile manufacturers and engineers by determining the best method and identifying the challenges.

The main contributions of the paper are summarized below.

- This review thoroughly examined the classification of conventional and advanced SOC estimation techniques.
- The estimation techniques were reviewed, focusing on the estimation algorithm, estimation error, advantages, and disadvantages.
- The various challenges, issues, and recommendations for monitoring SOC estimation were thoroughly discussed.
- Finally, the review provides valuable recommendations for developing an advanced BMS and efficient estimation methods for future sustainable EV applications.

The remaining paper is divided into six sections. Section 2 describes the framework of BMS. Section 3 describes the SOC estimation algorithms for estimating a battery SOC.

Section 4 describes the comparison of different SOC estimation methods. Section 5 describes the factors, challenges, and recommendations for a BMS. Finally, Section 6 depicts the concluding remarks.

## 2. Framework of BMS

Currently, a BMS is commonly employed by several vehicle companies, universities, and colleges. BMS goods have been advanced by several corporations such as EV Power Australia, the British REAP organization, American Edition Company, Beijing Significant Power Technology, and Harbin Guantuo Power Equipment Company [30]. The application of a BMS in EVs remains at the initial point. The base is that the quantity of batteries is 100 times above that of transportable devices in EVs [33–36]. Additionally, EVs are planned to supply high currents, voltage, and power. This process makes a BMS extra tricky compared to portable electronics.

The general role of a BMS is shown in Figure 2 [37], which shows the general function of a BMS, which consists of various kinds of actuators, sensors, signal lines, and controllers. The sample circuit measures temperature, voltage, and current, affording the gating sign achieved from the controller circuit. The vital work of the controller circuit is to estimate the SOC, state of health (SOH), state of energy (SOE), and state of power (SOP) of batteries over progressive algorithms and analog signals. The battery measurements of voltage, temperature, and currents are changed. After that, the data will be communicated to the vehicular controller and supply significant choice issues for vehicular and power distribution [38–41]. The BMS analyzes the EV power distribution and energy storage faults. Many researchers have proposed battery models in various ways. From [42], Figure 3 shows the BMS section divided into software and hardware assemblies.

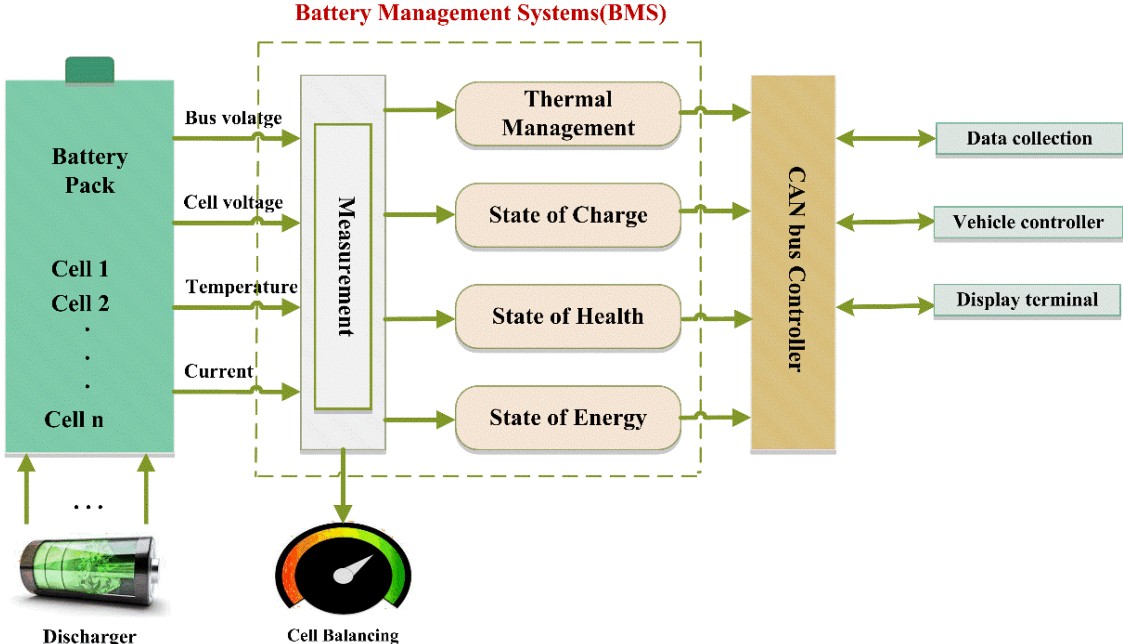

**Figure 2.** The general role of a BMS [37].

### 2.1. BMS Hardware

BMS uses various sensor frameworks to screen and measure battery parameters such as current, temperature, and voltage. Some researchers propose EIS (electrochemical impedance spectroscopy) to screen battery cell impedance [43]. High-cost devices and space limits make high-accuracy information outside the lab difficult to obtain. To stop overheating, charging, and discharging, a protection system must be developed. Constant voltage/current is used to charge batteries, and a galvanostat and potentiostat may be needed. Balance cells may also need a variable rheostat. Balancing cells is a key strategy

for improving battery pack stability and estimating battery life. Temperature affects cell reliability, performance, and imbalance. Thus, some authors [44] have acknowledged that reducing temperature differences between cells is important and should be observed and worked on. A BMS unit works independently after data/information transfer. A controlled transceiver is required to send data inside the BMS. With wireless telecommunication and smart batteries, the charger and battery can share a wealth of information [45].

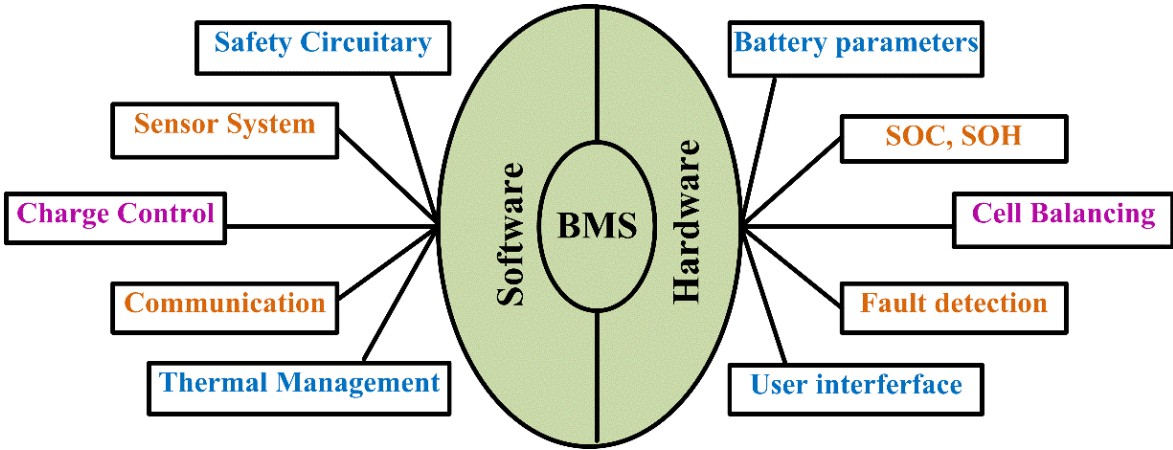

**Figure 3.** The basic outline of a BMS in an EV.

### 2.2. BMS Software

BMS software is the arrangement's midpoint. It controls sensor data and hardware operations to create choices and state approximations. BMS software must include a sample rate, switch control checking in the cell balancing controller, a sensor scheme, and a uniform active security circuit strategy. Online processing and research are required to inform and regulate battery functions. Robust automated information analysis and reliability may be key because the study handles state assessment and fault finding. This information will be presented to the operator in an easy-to-use interface. Below are BMS-specific roles. Total cell voltage, current, separate cell voltage measurement, temperature, impedance, and smoke detection are battery parameters.

Battery state estimation includes SOH and SOC, which group working situations supported by state–space representations, NN (neural networks), symbolic/fuzzy logic, etc. [46]. Cell balancing without over-discharging/charging maximizes battery performance. It aligns with SOC cell stages. The controller can control charging based on each cell's SOC. Thus, a precise estimation of the SOC of each cell is required to improve cell balancing. Online processing will expose sensitive issues. Data analysis is needed to determine battery faults and out-of-tolerance conditions. Before potential problems, important information will be noted. The BMS interface must display vital data. On the control panel, the battery SOC shows the range. Additionally, irregular, disturbing, and extra ideas are wanted to inform the operators of the battery estimate and calculation [47–49].

Figure 4 shows the BMS block diagram. The working detail is broken down. The battery's measurement block converts current, temperature, and voltage into digital signals at each point.

These constraints are used to evaluate the battery's SOH and SOC. A capability estimation block is used to control the max charging/discharging current. The cell balance block uses the capability estimation results to limit over-discharge/charge irregularities. Ground fault-finding improves system safety. The thermal management lump monitors the temperature to ensure battery safety. An input- and the output-controlled transceiver is used. To receive and transmit massive amounts of data, a high-speed, controlled transceiver is required.

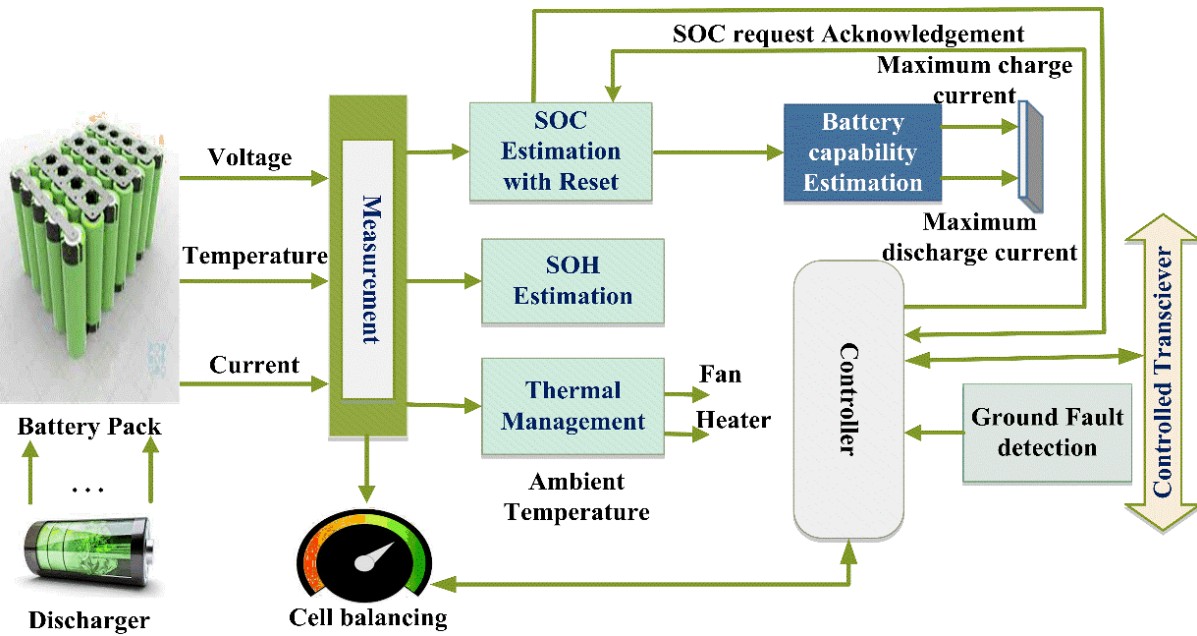

**Figure 4.** Block diagram of the BMS.

The various currently promoted BMS separately play out the elemental capacities in an unpredicted mode. Table 4 shows the comparative analyses of different BMS products.

**Table 4.** Comparative analyses among various BMS products.

| Parameters | Maxim DS2726 [50] | TI BQ78PL114 [51] | OZ890 [52] |
|---|---|---|---|
| Cell constraints measured | Voltage as well as current | Voltage, temperature, impedance, and current | Current and voltage |
| Pack constraints measured | Not available | Not available | temperature |
| Safety protection | • Short circuit current<br>• Over current<br>• Over voltage | • Three power field-effect transistors<br>• One secondary safety output fuse | • Short circuit current<br>• Over current |
| Estimation of SOH/SOC | None | SOC | SOC |
| Data logging | No | On PC-based GUI only | EEPROM |
| Dissipative equalization of cell | Charge shifting | Not available | External resistance stable |
| Communication | Unknown | Power LAN, SMBus | CAN |
| Non-dissipative equalization of cell | Not available | Inductive charge shuttle | Not available |

Disadvantages of the other referenced BMSs incorporate the following:

- Restricted information working performances: The knowledge of working function plays a crucial part in database formation and stores the driving design. It can support developing as well as updating the SOC model.
- Absence of SOH and SOC estimations: SOH and SOC are utilized to define the present health standing and, therefore, the outstanding practice of the battery that may ensure the reliable and planned support operation of the battery substitution.

**Apprehensions about today's BMS vehicles**

Because of the growing hydrocarbon charges and ongoing revolutions in the technology of batteries, HEVs and EVs were introduced in the early 1990s and typically developed in the 2000s. While the development of BMSs has been insufficient and slow in such cases, Li-ion batteries have been widely employed in the past decade for EVs because of their

favorable properties such as high efficiency, life cycle, and energy density. The recent developments in BMSs for Li-ion batteries in EVs are discussed in [53].

Figure 5 shows the overview of a few works of literature that studied different SOC estimation methods.

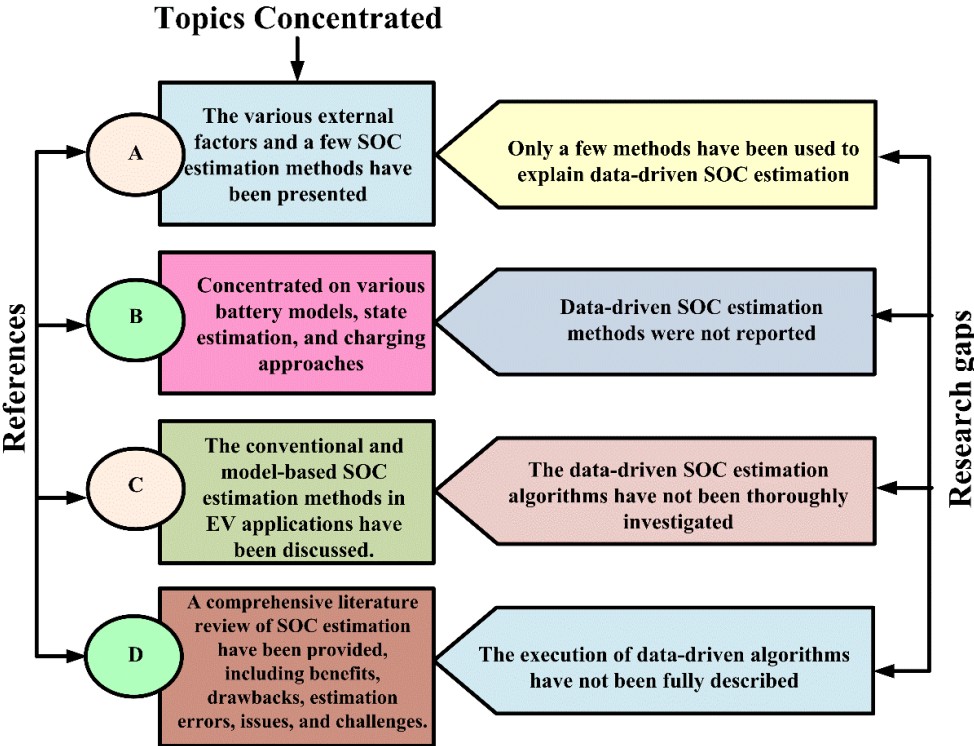

**Figure 5.** Overview of a few literature studies on different SOC estimation methods. "A" is referred to as [23], "B" is referred to as [34], "C" is referred to as [37] and "D" is referred to as [49].

## 3. State of Charge (SOC)

There has been a significant worry for all energy-storing devices for SOC estimation. SOC estimation gives us data and estimates the reliability of batteries with high precision. Since the 1980s, numerous methods have been introduced to estimate SOC. Although SOC estimation is a crucial challenge in EV batteries, it cannot be measured directly. It requires a specific algorithm for describing the battery's remaining capacity. The general architecture of the SOC system is shown in Figure 6. For SOC estimation, the current integration is the most traditional technique. The ratio of the available capacity to the battery's total capacity is shown in Equation (1).

$$SOC = 1 - \frac{\int i \, dt}{c_m} \tag{1}$$

where i indicates the current of the battery and $c_m$ indicates the total capacity.

The battery's total capacity decreases gradually due to its internal reaction and external load, leading to its nonlinear and non-stationary degradation characteristics. The categorization of SOC estimation methods is shown in Figure 7. Different kinds of literature have been presented in various manners. Every technique has its unique advantages along with disadvantages. In this review, SOC estimation methods were divided into five types: conventional, adaptive filter, learning algorithms, nonlinear observers, and others. Again, each process was classified into sub-methods [54], described as follows.

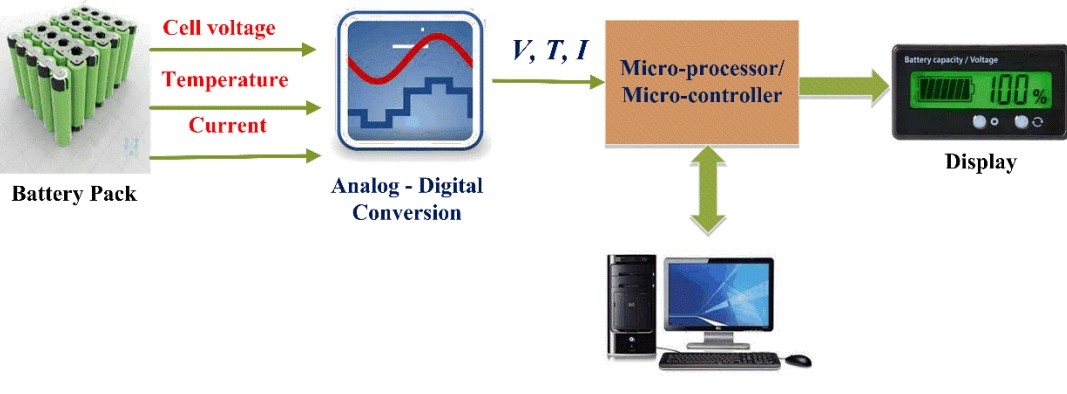

**Figure 6.** The general architecture of the SOC system.

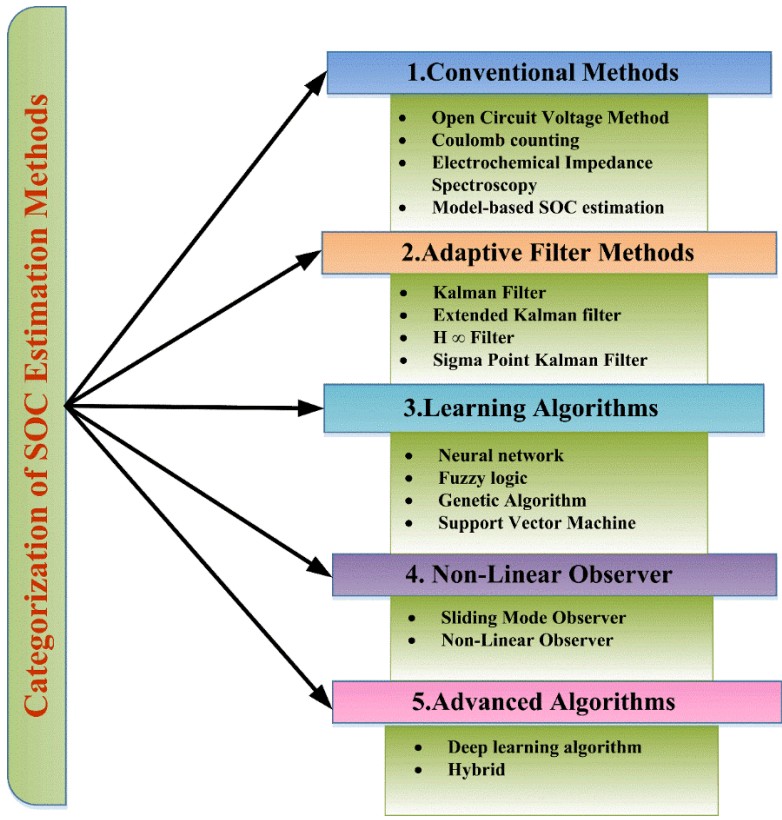

**Figure 7.** Categorization of methods for estimation of SOC.

*3.1. Conventional Methods*

3.1.1. Open-Circuit Voltage Method

The open-circuit voltage method has high accuracy, is easy to implement, and is a straightforward method, but its main disadvantage is that it takes more time to reach an equilibrium position. Therefore, online estimation of SOC is not an appropriate method. Therefore, this method is applicable only for low power consumption applications. Moreover, some observations are required to measure the discharge and charge voltage. For example, at high OCV, the battery is charged, and it is discharged at small OCV because of the hysteresis characteristics in batteries [55–58], as shown in Figure 8.

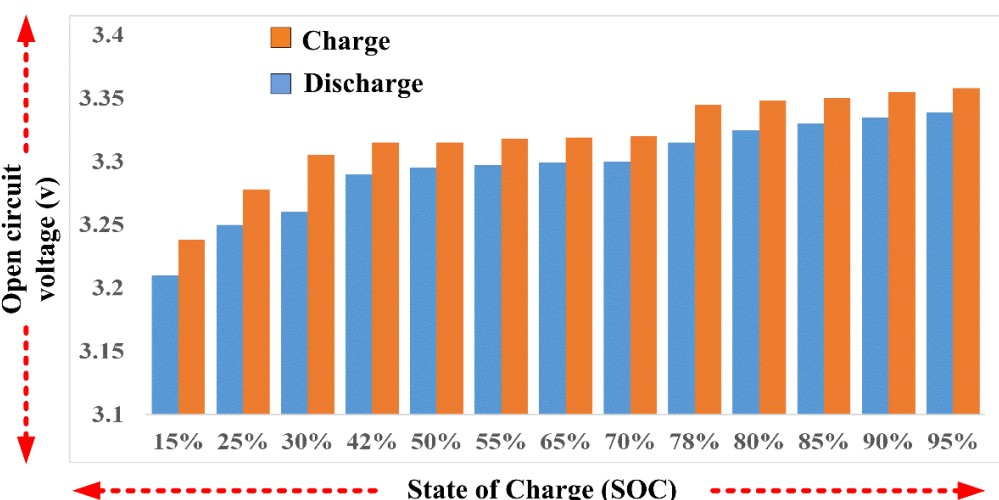

**Figure 8.** OCV vs. SOC was tested at 25 °C.

### 3.1.2. Coulomb Counting (CC) Method

The CC technique is the easiest one to estimate the SOC of the battery, and it can be implemented very quickly with low power calculation. The charging/discharging of the battery depends upon the integration of the current concerning time. It is expressed in Equation (2).

$$SOC = 1 - \frac{\int i.\eta dt}{c_m} \tag{2}$$

where η indicates the Coulombic efficiency, i indicates the current of the battery, and $c_m$ indicates the total capacity.

However, due to its uncertain disturbances, noise, temperature, and current, its results could be inaccurate. Furthermore, more difficulties exist in determining the SOC initial values, which might cause a cumulative error [59]. Additionally, to attain the maximum capacity, this method needs periodic capacity and complete cell discharge, which shortens the battery's lifespan [60].

### 3.1.3. Electrochemical Impedance Spectroscopy (EIS)

To implement the EIS, an appropriate electrochemical model is needed. Then, it evaluates the battery impedance by using capacitances and inductances over an extensive range of frequencies [61–63] that are recognized in an equivalent circuit model that includes two capacitive arcs and an inductive arc operated at low and high frequencies. Under various SOC values, a nonlinear LSF technique is utilized for computing the model impedances. If the system is not functioning in stable conditions, the EIS outcomes are difficult to reproduce. It has the advantages of low cost, operating online, and attaining good accuracy. From the actual values, the effect of battery temperature and the aging difference could vary the estimated outcomes, resulting in a deficiency of precision.

### 3.1.4. Model-Based SOC Estimation

Since the open-circuit voltage scheme cannot execute online, it needs appropriate rest time to monitor the SOC, which means it cannot be applied while the vehicle moves. Therefore, for online SOC, battery model development is essential. The most used battery models include the electrochemical [64–68] and equivalent circuit models [69,70]. An electrochemical model is used to study the battery's performance, which relates to the internal materials and considers the chemical thermodynamics and the electrodynamics effect. It can be expressed as:

$$V = VOC - VR - VP \tag{3}$$

where V denotes the terminal voltage, VOC means open-circuit voltage, VR indicates the potential difference across resistance, and VP represents the electric potential.

The RC networks have been used for the equivalent circuit model by considering dynamic and polarization characteristics. Using RLS (recursive least square) algorithm, the online OCV is executed and, for various RC networks, the outcome of the RLS algorithm is compared with experimental results. A model-based estimation is used online, and it has high precision. The drawback of this method for the specific battery is that a complete explanation of the electrochemical reactions is required, and it highly depends on the model's accuracy. Table 5 shows the analysis of the different conventional SOC estimation methods.

**Table 5.** Analysis of the different conventional SOC estimation methods.

| Technique | Pros | Cons |
|---|---|---|
| OCV [55–58] | <ul><li>Very simple method</li><li>High accuracy</li><li>Cost-effective</li><li>Easily implemented</li></ul> | <ul><li>Not suitable for online</li><li>Reaching an equilibrium state requires a long time.</li></ul> |
| CC [59,60] | <ul><li>Low power consumption</li><li>Easily implemented</li><li>Simple method</li></ul> | <ul><li>Inaccurate outcomes</li><li>Difficulties in defining the SOC initial values</li></ul> |
| EIS [61–63] | <ul><li>Operates online</li><li>Low cost</li><li>If the impedance value is stabilized, it attains good accuracy.</li></ul> | <ul><li>The effect of battery temperature and the aging difference could vary the estimated outcomes.</li></ul> |
| Model-based [64–71] | <ul><li>Operates online</li><li>High precision</li></ul> | <ul><li>It highly depends on the accuracy of the model.</li></ul> |

### *3.2. Adaptive Filter (AF) Algorithm*

3.2.1. Kalman Filter (KF) Algorithm

KF is a well-designed and intelligent tool commonly used in automobiles, navigator tracking, and aerospace applications. The striking feature of the Kalman filter is it has a self-correcting nature. A Kalman filter linear model contains a state equation, which predicts the current state, and a measurement equation, which updates the current state [72], which are expressed as follows:

$$\text{State equation} : x_{m+1} = A_m x_m + B_m u_m + f_m \tag{4}$$

$$\text{Measurement equation} : y_m = C_m x_m + D_m u_m + z_m \tag{5}$$

where A, B, C, and D represent the covariance matrices, x is the system state, f represents the process noise, u represents the control input, y represents the measurement input, and z represents measurement noise. Ting et al. [73] developed an RC battery model, which is used for modeling a Kalman filter. To explain the dynamic battery characteristics, the RC model mathematical equations remain converted into a state–space model to describe the dynamic battery characteristics. The outcome indicates that the estimated RMS error of the SOC using the Kalman filter is minor compared to the measured error. The authors of [74] also used the same method on the electrical equivalent model of a Li-ion battery with the help of the dSPACE real-time card and Matlab/Simulink software. The estimated SOC error was less than 5%. Yatsui [75] combined the results of a Kalman filter with two methods, the OCV, and the CC methods, to ameliorate the non-ideal factors. After executing the Kalman filter, the SOC precision was improved, with an error of ±1.76%. However, the Kalman filter cannot be used directly. It needs a complex calculation and is profoundly dependable with great strength to various working conditions and battery aging. On the other hand,

MI-UKF is impervious to unanticipated operational requirements and can improve UKF accuracy by more than 1% [76–78].

### 3.2.2. Extended Kalman Filter (EKF)

EKF has been applied to work the framework in nonlinear applications. It uses first-order Taylor series expansion and partial derivatives to linearize the battery model. At every instant of time, the state-space model is linearized and equates the predicted value of the battery with the measured voltage to precisely approximate the constraints for the SOC. If the scheme is exceptionally nonlinear, a linearization blunder might still happen. In any case, the linearization blunder could happen when the framework is profoundly nonlinear since the first-order Taylor series experiences an absence of precision in an exceptionally nonlinear state. Finally, the improved dual AEKF algorithm was applied, and the SOH and SOC estimation errors were within 1% [79–86].

### 3.2.3. H ∞ Filter

This is a very simple method in the designed model, and it does not have to know any details and measurement characteristics of noise. It considers only the time-varying parameters of the battery to carry out the system under the specific condition, which has robust strength. The precision of the model is deviated due to hysteresis, aging, and temperature effects [87–90]. In [91], this method was introduced to estimate battery SOC. The time-varying parameters are current, SOH, and temperature for second-order RC filter circuit design. An HPPC (hybrid pulse power characterization) experiment was performed to extract the voltage, resistance, and present characteristics. The projected model was tested using six Urban Dynamometer Driving Schedule tests and attained a good accuracy. In [92], the adaptive H∞ filter was introduced to estimate SOC. In this method, a polynomial function is helpful to evaluate the system functions, and the performance is examined and then compared with the adaptive extended Kalman filter (AEKF). The AHF performed better in accuracy and computational cost than other methods.

### 3.2.4. Sigma Point Kalman Filter

This is another nonlinear technique for the calculation of states, and it achieves more precise outcomes than the extended Kalman filter. The sigma-point Kalman filter (SPKF) algorithm is subjected to a numerical approximation. The algorithm selects sets of sigma points that are identical to the mean and covariance values of the developed model. The SPKF has the advantage of having a similar calculation. Furthermore, without taking Jacobian matrices into account, the complexity of the EKF is reduced [93,94]. The SPKF can demonstrate more accuracy while using less memory and performing fewer computational calculations. However, the estimated SOC was compared with the SPKF, Luenberger observer, and EKF algorithms, and the drawbacks are heavy and complicated calculations [95,96].

### *3.3. Learning Algorithms*

### 3.3.1. Neural Network (NN) Algorithm

An NN is a self-learning algorithm and also an intelligent tool. It uses trained data to estimate the state of the charge without knowing the initial data of the SOC. It consists of input, hidden, and output layers to form an NN structure, as shown in Figure 9 [97]. Building the NN structure takes discharge current, temperature, and voltage as inputs and the SOC as the output. The benefit of an NN is that it has a talent for being employed in nonlinear battery circumstances. The drawbacks are that training requires a large amount of data and a big memory to store the information [98,99]. Table 6 shows the analysis of the different adaptive filter SOC estimation methods.

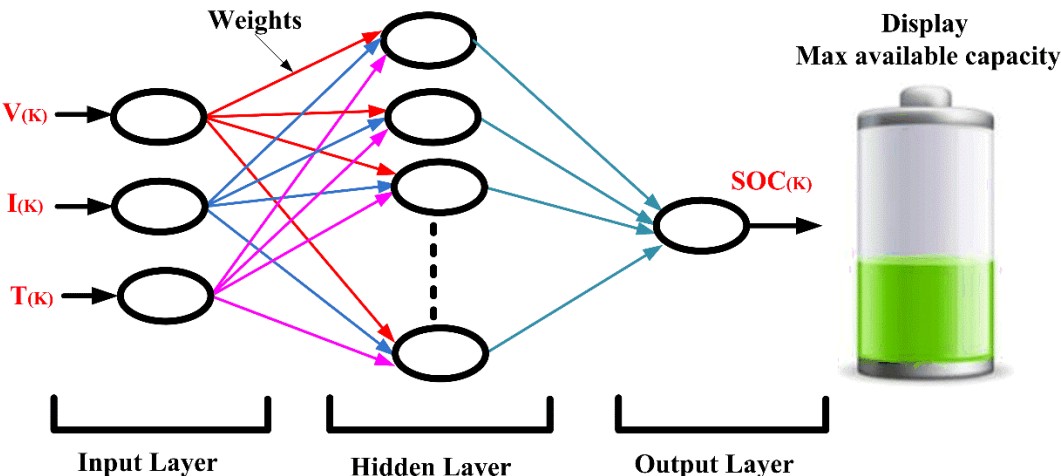

**Figure 9.** The comprehensive structure of neural network for SOC estimation [97].

**Table 6.** Analysis of the different adaptive filter SOC estimation methods.

| Technique | Pros | Cons |
|---|---|---|
| Kalman Filter [72–78] | • Self-correcting nature<br>• Intelligent tool<br>• Accurately estimates | • It cannot be used directly for SOC estimation.<br>• Needs complex calculations. |
| Extended Kalman Filter [79–86] | • Predicts nonlinear dynamic errors<br>• Improves accuracy | • Limited robustness<br>• Linearization error occurs |
| H ∞ Filter [87–92] | • Computational cost<br>• Satisfactory performance in precision | • Deviates the precision values due to aging and temperature |
| Sigma-Point Kalman Filter [93–96] | • Robustness<br>• Improvement in precision | • Heavy calculations<br>• Complicated |

### 3.3.2. Fuzzy Logic Algorithm

FL is the most influential algorithm for extending nonlinear, complex prototypes by using the training data. The employment of fuzzy logic includes rule-based inputs and outputs, a reasoning membership function, and defuzzification. However, estimating a nonlinear model is a powerful function. It needs an intricate calculation, dispensation unit, and large memory storage. Salkind et al. [46] applied FL for the estimation of SOC by using CC method data. This method uses three inputs at different frequencies, including impedances and SOC. It predicts the SOC with a max ±5% of error. The advanced ANFIS is most effective for estimating the SOC in Li-ion batteries; it was studied and applied in [100–105].

### 3.3.3. Genetic Algorithm (GA)

A genetic algorithm is mainly used for finding the optimum parameter. The primary function of a GA is to alter the constraint's trendy active method to improve the efficacy of the arrangement. It has been applied in mathematics, physics, and engineering for identifying nonlinear optimal parameters. Zheng et al. [106] used a genetic algorithm to assess four LiFePO4 battery cells, which were allied on a sequence. In addition, the outcome of this method was under 1% of the estimated SOC error. Xu et al. [107] applied a genetic algorithm for finding the parameters. By using various driving cycles, the method was validated; the outcome of this method had better accuracy, with below 1% error.

### 3.3.4. Support Vector Machine Algorithm

The SVM algorithm practices regression algorithm and works on kernel function, which is intended for converting the nonlinear type in an inferior measurement into a linear variety in an extreme measurement. In [108], the SVM technique was used for SOC estimation. The independent variables current, temperature, and voltage were obtained to excerpt the model constraints even though the batteries were discharging/charging. This method was authenticated, and an approved extreme SOC precision of 0.97 estimated quantity was determined. The benefits of SVM are performing in high-dimension models and nonlinear forms. By using training data, the SOC is estimated quickly and accurately. The drawback of this method is that trial and process errors are needed and require a long time [109]. Table 7 shows the analysis of the different learning SOC estimation algorithms.

**Table 7.** Analysis of the different learning SOC estimation algorithms.

| Technique | Pros | Cons |
|---|---|---|
| Neural Network [97,98] | • Accomplished work of batteries in nonlinear circumstances | • For storing the trained information, it needs a bulky memory unit. |
| Fuzzy Logic [100–105] | • Performs well<br>• It is very effective. | • Complex computation<br>• Large memory storage is needed.<br>• Costly |
| Genetic Algorithm [106,107] | • High accuracy<br>• Robust | • Heavy computation<br>• Good tuning parameters are needed to obtain effective outcomes. |
| Support Vector Machine [108,109] | • Performs outstandingly in nonlinear models<br>• Performs well in high-dimension models | • Heavy computation<br>• Requires trial along with process error to alter the parameters. |

### 3.4. Nonlinear Observer (NLO)

#### 3.4.1. Sliding Mode Observer (SMO)

SMO is an improved training controller for ensuring robustness and constancy of the system alongside model uncertainties as well as ecological disturbances. SMO is established by using the state equation in the next stage, which is decayed to the observer questions. In [110], a developed SMO was introduced to balance the nonlinear battery dynamic characteristics by using an RC circuit. This method can provide a controller for the conjunction period at the sophisticated discharge/charge value rate. The UDDS is situated to justify the method, and outcome details showed under 3% of the SOC error. In [111], the adaptive gain sliding mode observer (AGSMO) algorithm estimated the battery SOC on a combined equivalent circuit model. To extract the constraints, a battery pulse was used, and, by using the circuit model as well as terminal voltage, the state equations were developed. Experiments were performed to assess the recommended archetype, and outcomes proved that the model has an advantage in regulating the toughness derived when affecting all sound-on wrinkles.

#### 3.4.2. Nonlinear Observer (NLO)

Several observers have been applied, including both a linear observer [112–114] as well as a nonlinear [115] observer, to estimate the SOC. In [116], the NLO-dependent SOC estimation was introduced via a first-order corresponding RC circuit. This model was performed by using a driving cycle as well as a discharge test, and the outcomes were improved compared to extended KF and SMO in standings of speed and precision as well as cost. They are still discovering an appropriate gain matrix to decrease the error. Table 8 shows the analysis of the different nonlinear observer SOC estimation algorithms.

**Table 8.** Analysis of the different nonlinear observer SOC estimation algorithms.

| Technique | Pros | Cons |
|---|---|---|
| Sliding Mode Observer [110,111] | • Robustness<br>• It enhances sstability. | • Difficult to alter switching gain |
| Nonlinear Observer [112–116] | • Improved accuracy<br>• Improved convergence speed<br>• Robustness | • To reduce the error, it is problematic to find the appropriate gain matrix. |

*3.5. Advanced SOC Estimation Techniques*

3.5.1. Deep Learning Algorithm (DLA)

Deep learning (DL) algorithms have contributed to a better understanding of SOC estimation. Among the most notable are the long short-term memory (LSTM) network, deep neural networks (DNN), gated recurrent unit (GRU), and convolutional neural networks (CNN). The LSTM network [117] provides a strong SOC estimation performance because of its strong self-learning ability. The SOC of a battery is estimated using an LSTM network based on measured voltage, current, and temperature. Furthermore, DNN [118] exploits the battery's dependent behaviors on ambient temperatures and encodes them into DNN weights, resulting in a competitive estimation performance over a wide range of temperatures. GRU [119] is used to estimate the battery SOC at different temperatures and to evaluate the performance of two common lithium-ion batteries. Unlike a traditional feedforward neural network, the RNN employs hidden nodes to store information about previous inputs, allowing the SOC estimation to incorporate this information. LSTM and GRU are RNN variants that extend the original RNN's ability for long-term dependency. Another successful architecture in deep learning research is CNN. While the LSTM defines long-term dependency and is capable of handling time series data, the CNN employs convolutional behavior in a certain way to extract interconnections among input data. To model the complex battery dynamics, a combined CNN–LSTM network was proposed [120]. The CNN was specifically used to obtain advanced spatial features from the original data, while the LSTM was used to model relationships between the current SOC and past and present inputs. Both CNN and LSTM networks capture both spatial and temporal features of battery data. Table 9 shows the analysis of the different deep learning SOC estimation algorithms.

**Table 9.** Analysis of the different deep learning SOC estimation algorithms.

| Technique | Pros | Cons |
|---|---|---|
| LSTM [117] | • Has a track record of success in the face of long-term dependencies.<br>• During the online stage, computation is less intensive. | • Complex training execution requires the use of an expensive device to enhance training. |
| GRU [118] | • Long-term sequential dependencies are captured.<br>• LSTM gating mechanism issues are addressed. | • It necessitates a large amount of training data as well as a large storage device. |
| CNN–LSMT [120] | • Improved tracking precision<br>• It has a strong nonlinear fitting ability. | • The structure is complex, with a hidden layer and a visible layer. |

3.5.2. Hybrid Methodologies

The mixing of two or more algorithms is known as a hybrid, which improves the accuracy and efficiency of the battery. It requires a large memory unit because of its complex mathematical computations. However, a hybrid methodology accomplishes consistent as well as operative outcomes and then, likewise, decreases the BMS price. In [121], the multi-state and extended Kalman filter methods were proposed, using the equivalent

circuit model. The prototype is situated to move for a discrete state space that can provide supplementary data as opposed to linearized data by utilizing a Jacobian matrix. The simulation outcomes provided better accuracy, with a 2.7% average error. In [122], the CC, KF, and OCV methods were reviewed for SOC estimation. First, by using the OCV and CC methods, the SOC was estimated, which decreased the estimated error of CC. Then, the Kalman filter was utilized to enhance the precision value of the SOC.

In [123], a hybrid methodology was introduced; it included the CC and EKF methods for a time-changing dynamic estimation. The first open-circuit voltage method was applied for the SOC. The EKF was applied for the corrected SOC values, and this process was continued until the battery was fully discharged. The accuracy of the model was under 6.5%.

In [124], SOC was estimated based on AUKF utilizing RBF, and it was utilized to alter the particulars of the system. The AUKF stayed employed for evaluating the SOC. Then, united, both methods were equated by adaptive KF. The results of the AUKF were superior to the adaptive KF from the perspective of error. In [125], the H $\infty$ filter and discrete-time KF were applied to the nonlinear model of the Li-ion battery. The outcomes of this method were compared with adaptive Luenberger as well as SMO-based estimation models, and the accuracy of this method was improved, with <1% of error. In [126], the SOC of the lithium-ion cell was adaptively estimated using the multiple model adaptive estimation (MMAE) technique using a modified enhanced self-correcting (ESC) cell model. When compared to the EKF result, the SOC estimation converged more quickly. In [127], this study designed an enhanced Kalman filter (KF)-based adaptive observer by approximating the electrochemical model. The estimator's predictions were compared against the experimental data in simulations. The simulation outcomes were more precise and efficient than those of the KF. The accuracy of this method was improved, with <2% of error.

In [128], EKF paired with an adaptive neuro-fuzzy inference system (ANFIS) reduced error and improved accuracy over EKF alone. The root mean square error (RMSE) compared the EKF with the EKF-assisted ANFIS. In this way, the hybrid technology improved precision and accuracy while reducing expenses. In [129], for Li-ion batteries with uncertain noise circumstances, a new noise adaptive moving horizon estimating (NAMHE) approach was suggested. The simulation outcomes showed that the suggested technique reduced the SOC estimate error compared to the classic moving horizon estimating (MHE) method. The RMSE of the suggested technique and MHE were 0.7543% and 1.3026%, respectively. In [130], different OCV test methodologies impacted the correlation of the OCV and SOC; an effective OCV–SOC relationship may increase SOC online convergence speed and accuracy. The AEKF SOC estimate technique was more accurate and reliable than EKF during driving cycles, with a 0.5481% mean error of the proposed system. Hybrid methods give accurate outcomes and are cost effective. Table 10 shows the analysis of hybrid SOC estimation algorithms.

**Table 10.** Analysis of hybrid SOC estimation algorithms.

| Technique | Pros | Cons |
|---|---|---|
| CC and KF [122] | • CC has low power consumption.<br>• KF has a self-correcting nature and is an intelligent tool. | • Inaccurate outcomes by CC and KF need complex calculations. |
| EKF and multi-state [121] | • Predicts nonlinear dynamic errors.<br>• Improves accuracy. | • Limited robustness<br>• Linearization error occurs. |
| H $\infty$ filter and discrete-time KF [125] | • High accuracy<br>• Robustness<br>• It enhances the stability. | • Deviates the precision values due to aging and temperature. |

**Table 10.** *Cont.*

| Technique | Pros | Cons |
|---|---|---|
| NAMHE [129] | • Worked in conditions with unknown noise levels. <br> • More stability and precision | • The program's computational complexity and memory use will rise as it runs. |

## 4. Comparisons

A comparison of different SOC methods is shown in Table 11. The table consists of different types: conventional methods, adaptive filter, learning algorithms, nonlinear observers, and hybrid. Again, each type can be subdivided into different methods and algorithms. First, the conventional methodologies [131–134] use a battery's physical properties involving resistance, impedance, voltage, and discharge current. The methods are very simple, very cost effective, and have high accuracy. Compared to the remaining methods, the average error is between $\leq\pm4$ to $\leq\pm5\%$. It is moderate, as shown in Figure 10. Second, the adaptive filter methodologies [75,91,135,136] use different algorithms and models to estimate the SOC; these methods have a self-correcting nature, are intelligent tools, and their estimates are but need complex calculations. The average accuracy error is $\leq\pm1$ to $\leq\pm2.19\%$, which is very low; the best method for SOC estimation is as shown in Figure 11. Third, learning methodologies [46,106,137,138] need heavy computation as well as a large amount of training data to define the Li-ion nonlinear characteristics for SOC estimation. Nevertheless, the advantages are efficiency, high accuracy, and robustness, but the average error is between $\leq\pm2$ to $\leq\pm6\%$, which is very high compared to filter algorithms, as shown in Figure 12. Fourth, nonlinear observer methodologies [110,131] are handled with highly nonlinear schemes, and the benefits are robustness and enhanced stability. However, the drawback is that it is difficult to alter the switching gain. The average error is between $\leq\pm3$ to $\leq\pm4.5\%$, higher than KF; they are less related to a conventional methodology, as shown in Figure 13. Fifth, in the deep learning [117–120] and hybrid methodology [121,123,139], the average accuracy error of deep learning is $\leq\pm1.33$ to $\leq\pm1.88\%$, as shown in Figure 14. The mixing of two or more algorithms is known as a hybrid, which improves the accuracy and efficiency of the battery. It requires a large memory unit because of its complex mathematical computations, and the average error is between $\leq\pm2.7$ to $\leq\pm6.5\%$, as shown in Figure 15. Table 12 summarizes the different SOC estimation methods for Li-ion batteries.

**Table 11.** Comparisons of average error (%) on different SOC estimation methods.

| Type | Methodology | Average Error (%) | Application in EVs |
|---|---|---|---|
| Conventional Method | OCV [131] | Unspecified | No |
| | CC [132] | $\leq\pm4$ | Yes |
| | EIS [133] | Unspecified | No |
| | Model-based [134] | $\leq\pm5$ | Yes |
| Adaptive Filter | KF [75] | $\leq\pm1.76$ | Yes |
| | EKF [135] | $\leq\pm1$ | Yes |
| | H ∞ F [91] | $\leq\pm2.49$ | Yes |
| | SPKF [136] | $\leq\pm2$ | Yes |
| Learning Algorithms | NN [137] | $\leq\pm4.6$ | Yes |
| | FL [46] | $\leq\pm5$ | Yes |
| | GA [106] | $\leq\pm2$ | Yes |
| | SVM [138] | $\leq\pm6$ | Yes |
| Nonlinear Observer | SMO [110] | $\leq\pm3$ | Yes |
| | NLO [125] | $\leq\pm4.5$ | Yes |

**Table 11.** *Cont.*

| Type | Methodology | Average Error (%) | Application in EVs |
|---|---|---|---|
| Deep Learning Algorithms | LSTM [117] | $\leq \pm 1.40$ | Yes |
| | GRU [119] | $\leq \pm 1.33$ | Yes |
| | CNN [120] | $\leq \pm 1.88$ | Yes |
| Hybrid | Hybrid [121] | $\leq \pm 2.7$ | Yes |
| | Hybrid [123] | $\leq \pm 6.5$ | Yes |
| | Hybrid [139] | $\leq \pm 3.5$ | Yes |

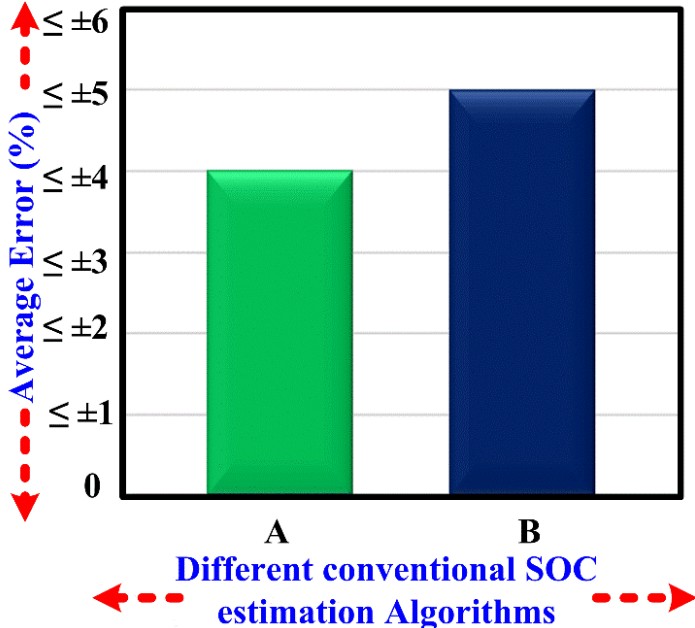

**Figure 10.** Comparison between the different conventional SOC estimation methods. "A" is referred to as [59], and "B" is referred to as [135].

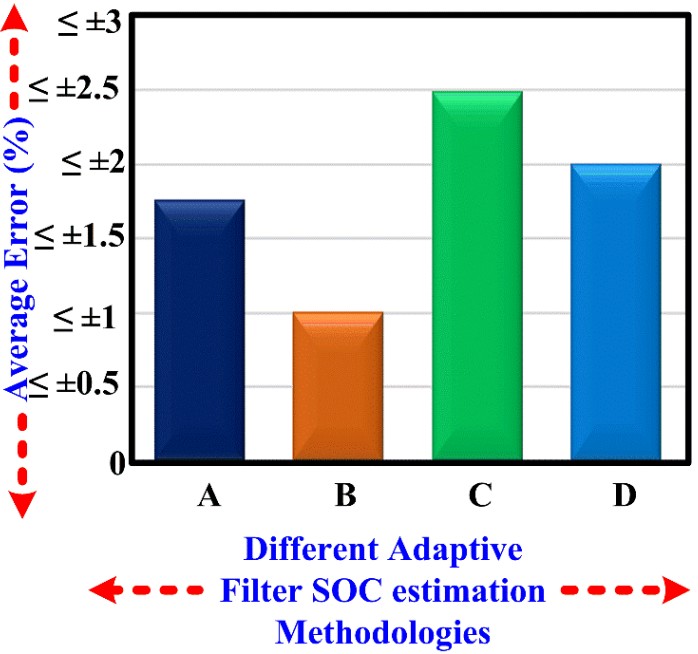

**Figure 11.** Comparison between the different adaptive filter SOC estimation methods. "A" is referred to as [75], "B" is referred to as [135], "C" is referred to as [91] and "D" is referred to as [136].

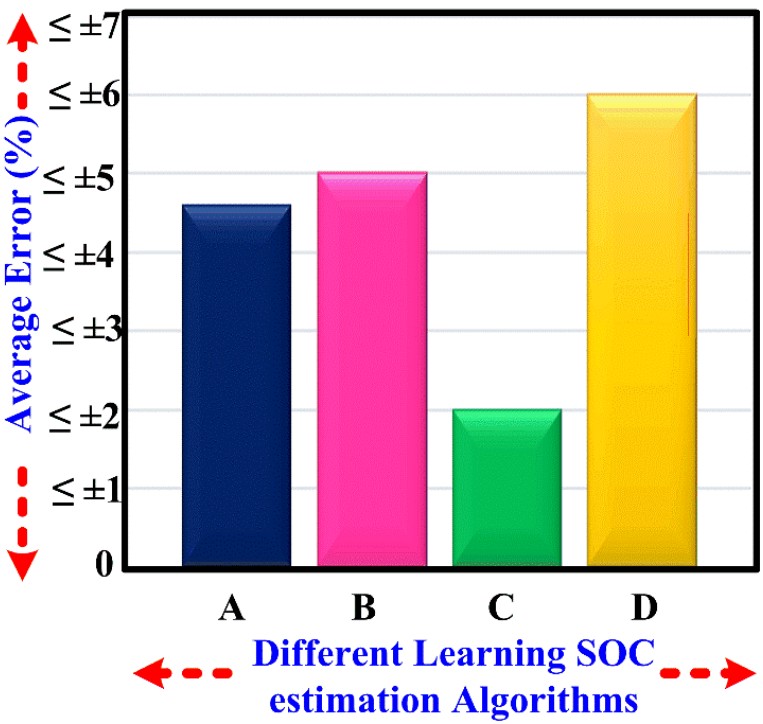

**Figure 12.** Comparison between the different learning SOC estimation algorithms. "A" is referred to as [137], "B" is referred to as [46], "C" is referred to as [106] and "D" is referred to as [138].

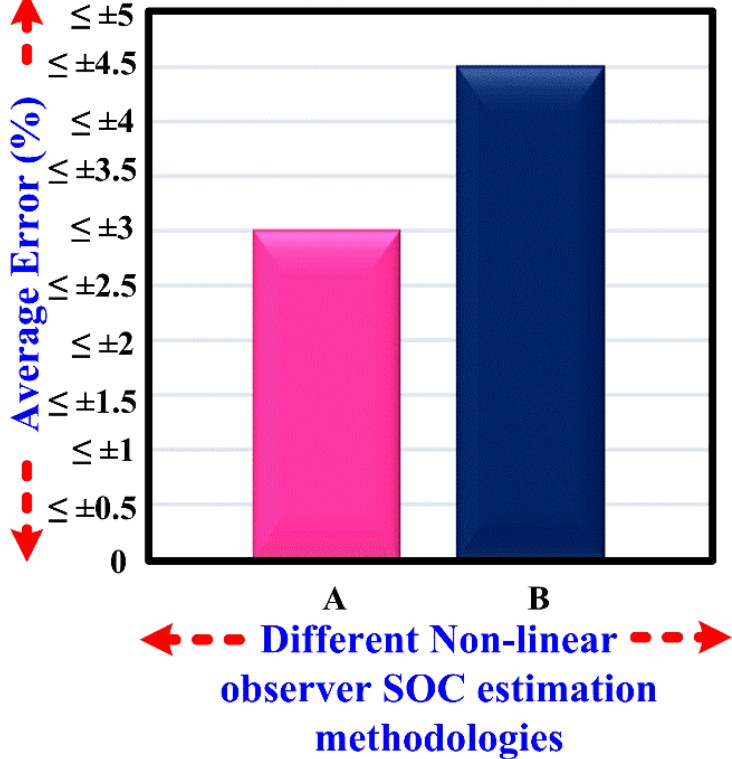

**Figure 13.** Comparison between the different nonlinear observer SOC estimation methods. "A" is referred to as [110], and "B" is referred to as [131].

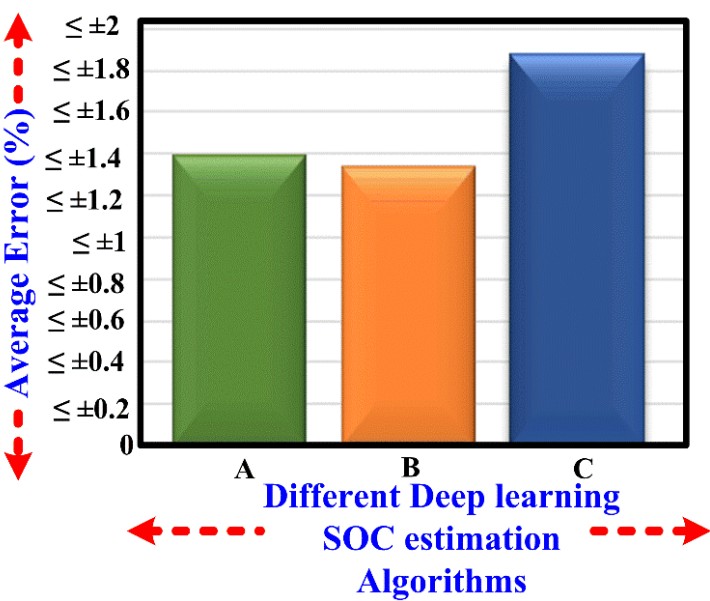

**Figure 14.** Comparison between the different deep learning SOC estimation algorithms. "A" is referred to as [117], "B" is referred to as [119], and "C" is referred to as [120].

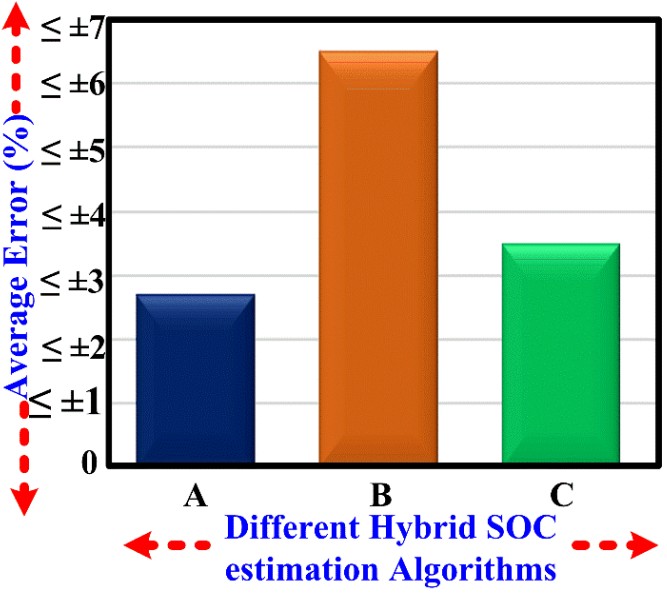

**Figure 15.** Comparison between the different hybrid SOC estimation algorithms. "A" is referred to as [131], "B" is referred to as [123], and "C" is referred to as [139].

**Table 12.** Summary of the different SOC estimation methods for Li-ion batteries.

| Type | Major Benefits | Major Limitations |
|---|---|---|
| Conventional Method [132–134] | <ul><li>Easy implementation</li><li>Low power consumption</li><li>High accuracy</li><li>Easy to understand</li></ul> | <ul><li>Not suitable for online</li><li>It highly depends on the accuracy of the model.</li><li>Susceptible to aging and temperature</li></ul> |
| Adaptive Filter [75,91,135,136] | <ul><li>High accuracy</li><li>Excellent filtering effect</li><li>Insensitive to initial SOC</li><li>High robustness</li></ul> | <ul><li>High computation complexity</li><li>Large computation cost</li><li>Unfit for large noise measurement</li></ul> |

**Table 12.** *Cont.*

| Type | Major Benefits | Major Limitations |
|---|---|---|
| Learning Algorithms [46,106,137,138] | • Independent models<br>• Great accuracy<br>• Rule-based inference<br>• Nonlinear mapping ability | • Requires a large amount of training data<br>• Requires large memory units<br>• Costly processing unit<br>• Time-consuming process |
| Nonlinear Observer [110,131] | • Robustness<br>• Powerful tracking performance<br>• The excellent nonlinear processing capability | • Inaccurate<br>• Difficult to find proper gain matrix<br>• Insufficient stability |
| Hybrid [121,123,139] | • More effective<br>• Reliable<br>• High precision | • Requires longer computation time<br>• High complex computation |

## 5. Factors, Challenges, and Recommendations

Due to performance degradation and complex electrochemical reactions, developing and arranging a Li-ion battery organization system for EVs is a top priority. Moreover, most well-defined battery experiments are performed in a controlled laboratory environment with constant current, voltage, and temperature. Few analyses exist on battery performance in severe, hot, wet, and rainy conditions. External mass affects battery capacity. Unmodeled consequences add to unconsidered algorithms and models. Temperature, aging, cell unbalancing, hysteresis characteristics, battery modeling, self-discharge, charge/discharge rate, etc. are also factors in battery performance decline. The work summarizes the key findings by applying aging modeling to four different Li-ion battery capacity loss datasets [58]. Figure 16 shows Li-ion battery cycle life versus temperature at different charge rates. Many researchers have proposed battery SOC models. Every model suffers from missing data for real-world EV applications. To accurately estimate battery states, complex calculations, high cost, and accuracy are issues. Table 13 lists SOC monitoring challenges, causes, and recommendations. Figure 17 explains battery anode aging.

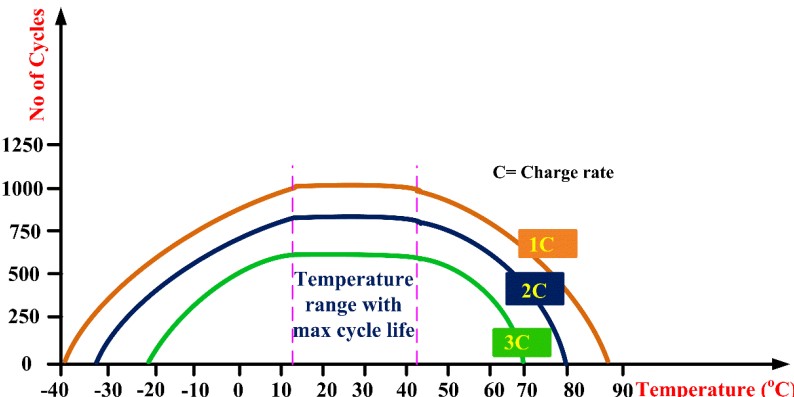

**Figure 16.** Battery cycle life vs. temperature at a dissimilar charge rate of Li-ion battery [58].

**Table 13.** Challenges, causes, and recommendations to monitor the SOC.

| Ref. No. | Challenges | Causes | Recommendations |
|---|---|---|---|
| [140–144] | Temperature | • It is caused by an increase in inconsistency and an increment in the electrolyte's development, which can support the movement consequence and particle diffusion. | • The finest temperature range and battery cycle charging rate are acknowledged in [144]. |

**Table 13.** *Cont.*

| Ref. No. | Challenges | Causes | Recommendations |
|---|---|---|---|
| [145,146] | Aging | • It is caused by capacitance degradation as well as internal resistance. | • An OCV curve model to evaluate the battery SOH is planned by enhancing one only constraint, the aging of batteries [146]. |
| [147–156] | Cell unbalancing | • Due to the manufacturing and chemical characteristics of the battery, which might vary while discharging and charging | • An active cell balancing mechanism is separated into two types, passive and active, which were proposed in [153]. |
| [156–160] | Hysteresis characteristics | • Ohmic resistance, electro-chemical issues, and concentration polarization are the key causing issues along with it being produced by scattering of energy in the development. | • Hysteresis assessment of Li-ion cells is established for improving the precision in contradiction of the impact of hysteresis [156]. |
| [161–164] | Battery modelling | • Due to the complex dynamics and electro-chemical environment, it is challenging to create a battery model. | • ESC model along with higher-order RC model was proposed in [70]. |
| [165,166] | Self-discharge | • Lithium species loss and SEI formation are accountable for causing self-discharge. | • ECN model for estimation of SOC by using prediction error minimization method was proposed in [166]. |
| [167–170] | Charge and discharge rate | • Phase dispersion is the key warning factor for high discharge current in plastic Li-ion batteries. | • The discharge as well charges in a current range of the Li-ion battery were acknowledged in [170]. |
| [171–174] | Communication method | • Due to the non-uniform charging mechanism, developing an advanced, even charger is problematic. | • Wireless expertise was employed to transfer the data between charger and battery in [173]. |

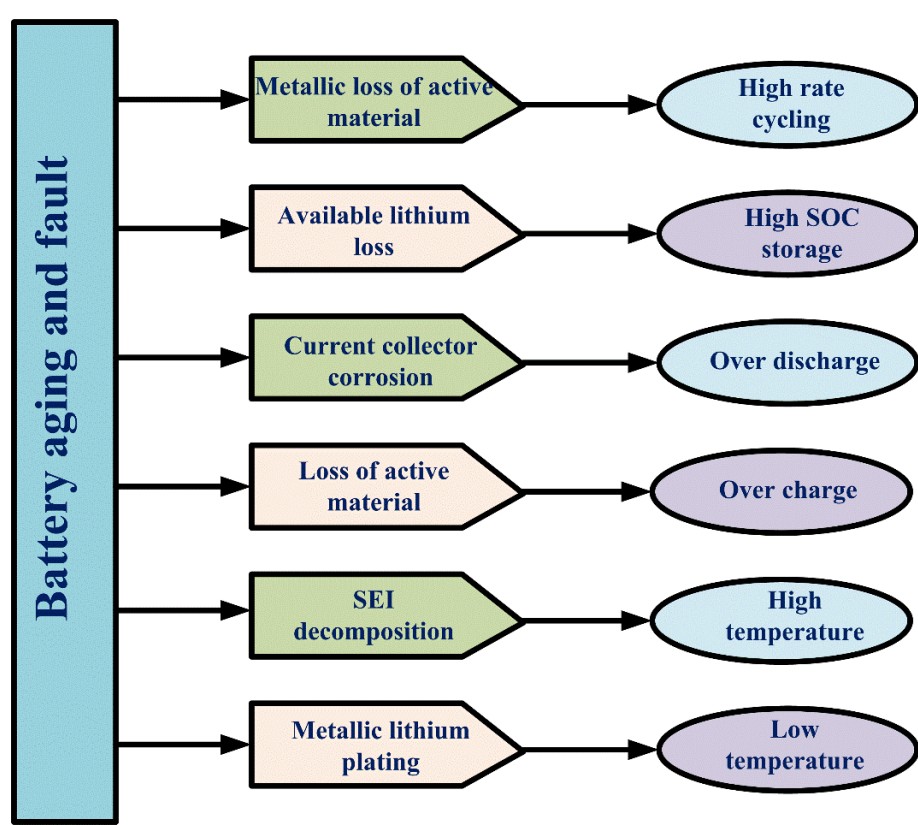

**Figure 17.** Explanations for the aging of a battery at the anode [145].

## 6. Conclusions

This paper critically reviewed BMS with attention to several methods for estimating the SOC. A Li-ion battery is recommended for complex in-vehicle operation due to having benefits such as high efficiency, energy density, voltage-generating capability, and long-life cycle span. The importance of a BMS to achieve reliable and safe operating of Li-ion batteries was described in detail. A BMS includes both hardware and software, which were discussed briefly. This analysis mainly explored several algorithms in addition to estimation methods of SOC. From the various literature reviewed, a complete explanation including method drawbacks, benefits, and estimation errors was broadly studied.

This review paper recognizes that the conventional techniques are simple, and implementation is also easy. Nevertheless, they are highly affected by temperature, aging, and external disturbances. Similarly, the situation was observed that an AF algorithm can calculate the nonlinear dynamic condition utilizing better accuracy, high efficiency, and low computational cost. Nevertheless, this method suffers from poor robustness and a heavy burden. Regarding the learning algorithm (LA), it executes a nonlinear dynamic modeling arrangement better by bearing in mind the temperature, aging, and noises. However, it requires composite computation and large memory storage parts. The nonlinear observer (NLO) method has improved robustness, accuracy, computation costs, and coverage speed. However, this method could provide inaccurate outcomes if the device is not designed correctly. Estimating a precision SOC has become a significant challenge due to the electrochemical reactions of several external and internal factors of Li-ion batteries.

## 7. Future Scope

Based on a comprehensive review of existing analyses on SOC estimation, this review makes several important recommendations for future research, as shown in Figure 18.

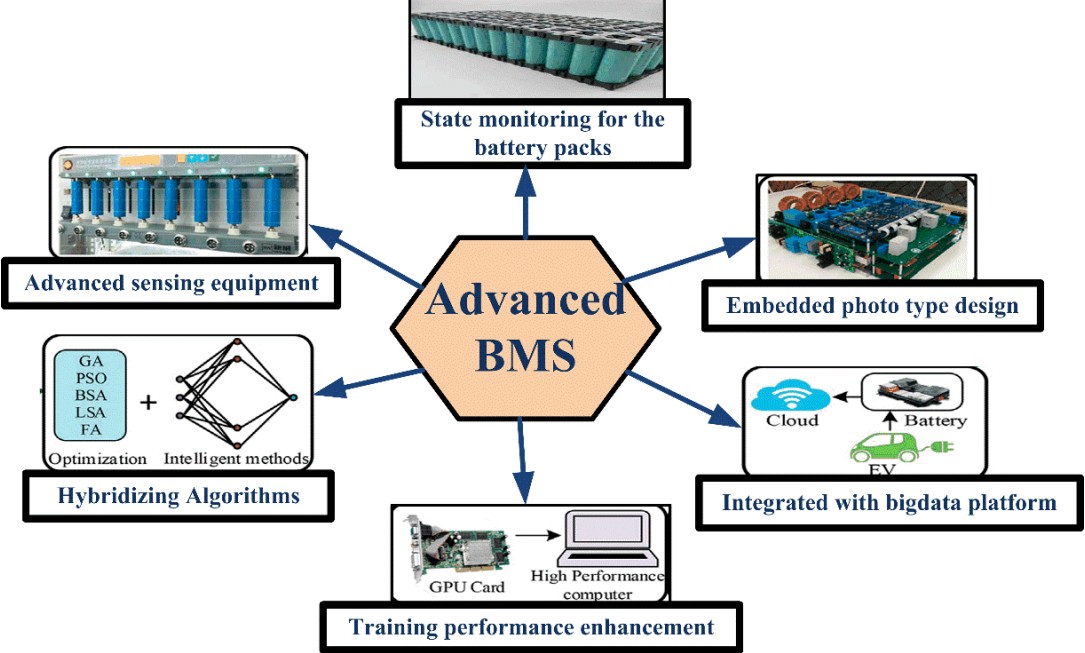

**Figure 18.** Future trends in advanced BMS for EV applications.

- **Hybridizing algorithms:** To achieve satisfactory SOC estimation performance, hybrid methods are highly recommended, in which multiple methods enhance each other.
- **Advanced sensing equipment:** It is essential in developing high-precision sensors to improve current and voltage measurement accuracy for accurate SOC estimation.

- **Cloud computing technology:** The real-time operation of intelligent algorithms and BMS controller schemes can be enhanced further with proper monitoring and analysis via the cloud storage and big data platform.
- **Embedded systems:** Additional research is needed to create an embedded prototype with a low computational cost and small memory units.
- **High-performance processors:** To accelerate the training operation, a GPU-based high-performance processor and appropriate activation functions, excitable parameters, and training algorithms are necessary.
- **State monitoring for the battery packs**: State estimation and fault diagnosis for battery packs must be evaluated to reduce cost, power loss, size, and voltage stress, and improve equalization time and efficiency.
- It is necessary to have a generalized validation and benchmark method for SOC estimation.

In conclusion, key information and the critical analysis obtained from this review will be useful for automobile engineers and the EV-related industries to develop and implement advanced BMSs for EV applications. Thus, further research on BMSs using advanced intelligent algorithms will improve battery performance and lifespan and ensure the safe and reliable operation of EVs, resulting in significant growth of the battery and EV markets. Furthermore, the battery-related market and expansion of the EV market can help achieve long-term development goals such as emission reduction, clean energy, economic development, and job creation. As a result, long-term future innovation is required to improve EV performance in terms of accurate battery monitoring and control strategy development, global collaboration, and sustainable development.

**Author Contributions:** Conceptualization and methodology, G.T.; investigation, D.C.; resources, D.C.; data curation, G.T.; writing—original draft preparation, G.T.; writing—review & editing, G.T.; visualization, G.T.; and D.C.; supervision, D.C.; project administration, D.C.; funding acquisition, D.C. All authors have read and agreed to the published version of the manuscript.

**Funding:** This research was funded by the Vellore institute of technology in Vellore, India, for conceding a SEED fund/12439 to subsidize this research.

**Data Availability Statement:** This review has no information related to it.

**Acknowledgments:** The authors are thankful to the Vellore institute of technology in Vellore, India, for conceding a SEED fund/12439 to subsidize this research. The authors, likewise, wish to thank the particular copyright holders for allowing approval to use the pictures, graphics, tables, and figures in this work.

**Conflicts of Interest:** The authors declare no conflict of interest.

## Abbreviations

The following abbreviations are used in this manuscript:

| | |
|---|---|
| AF | Adaptive Filter |
| AEKFANFIS | Adaptive Extended Kalman filterAdaptive Neuro Fuzzy Inference System |
| BMS | Battery Management System |
| CAN | Controller Area Network |
| CC | Coulomb Counting |
| CNN | Convolutional Neural Networks |
| DNN | Deep Neural Networks |
| EIS | Electrochemical Impedance Spectroscopy |
| EKF | Extended Kalman Filter |
| EV | Electric Vehicle |
| FL | Fuzzy Logic |
| GA | Genetic Algorithm |
| GHG | Greenhouse Gas |
| GRU | Gated Recurrent Unit |
| KF | Kalman Filter |

| ESC | Enhanced Self-Correcting |
| LA | Learning Algorithms |
| LSTM | Long Short-Term Memory |
| LCO | Lithium Cobalt Oxide |
| LTO | Lithium Titanium Oxide |
| LNO | Lithium Nickel Oxide |
| LFP | Lithium Iron Phosphate |
| LMOMHE | Lithium Manganese OxideMoving Horizon Estimation |
| MMAENAMHE | Multiple Model Adaptive EstimationNoise Adaptive Moving Horizon Estimation |
| NCA | Lithium Nickel Cobalt Aluminum Oxide |
| NMC | Lithium Nickel Manganese Cobalt Oxide |
| NLO | Nonlinear Observer |
| NN | Neural Network |
| OCVRMSE | Open-Circuit VoltageRoot Mean Square Error |
| SMO | Sliding Mode Observer |
| SOC | State of Charge |
| SOE | State of Energy |
| SOH | State of Health |
| SOP | State of Power |
| SPKF | Sigma-Point Kalman Filter |
| SVM | Support Vector Machine |

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
