# Peer review of "A Review on Different State of Battery Charge Estimation Techniques and Management Systems for EV Applications"

_electronics, doi:10.3390/electronics11111795_

Round 1
Reviewer 1 Report
This paper reviews the State-of-Charge (SOC) estimation techniques of the battery and battery management system for EV applications. Following are the reviewer’s comments:
- The state-of-art is done in this review paper is far from enough.
- A potential plagiarism issue is detected.
- Figure 4 referenced from [43] is different from the given reference [43].
- Figure 5 is exactly the same as the reference [50].
- The Fugue 6 is almost the same as the reference [50].
- Figure 7 is almost the same as the reference [50].
- Figure 8 is almost the same as the reference [50] and is very different from the given reference [59].
- Figure 9 is almost the same as the given reference [98].
- Similar work had been done intensively and NO new technical significance or point of view is provided in this paper.
Overall, similar review work had been done extensively but this paper fails to provide a new perspective. Furthermore, most of the work done in this paper is copied from others, causing the potential plagiarism issue.
Author Response
To
The Managing Editor,
Electronics,
MDPI.
Dear Editor,
Sub.: Submission of revision of paper electronics-1703990, A Review on Different State of Battery Charge Estimation techniques and Management System for EV Applications –Reg.
We would like to thank the Managing Editor, Handling Editor and reviewers for their valuable decision and recommendations made to improve the quality of the paper. We are happy to submit the revised manuscript entitled “A Review on Different State of Battery Charge Estimation techniques and Management System for EV Applications”, which we would like to submit for publication as a review article in Electronics. We thank you for providing such an opportunity, and we also thank the anonymous reviewers for their valuable and helpful comments. Based on the set of reviews the manuscript is thoroughly revised and the appropriate corrections have been made. The point-by-point response to the reviewers’ comments is included here forth for your kind perusal.
Thank you. We look forward to your positive response.
Yours sincerely,
Dr. Dhanamjayulu C
Assistant Professor (Senior)
School of Electrical Engineering,
Vellore Institute of Technology,
Vellore-632014. Tamilnadu, India.
E-mail: [email protected]
Author Responses:
We are very thank full to the editor and referee for the valuable time spent in reviewing this manuscript. We also thank them for their suggestions and comments to improve the manuscript. We have addressed all issues in the review report and believe that the revised version can meet the journal publication requirements.
Authors’ Responses to Reviewers’ Comments:
The authors, first of all, thank the reviewers for their valuable comments and suggestions.
RESPONSE TO REVIEWER’S COMMENTS - REVIEWER #1
This paper reviews the State-of-Charge (SOC) estimation techniques of the battery and battery management system for EV applications. Following are the reviewer’s comments: |
|
RESPONSE: Respected Reviewer, Thank you for your suggestions and comments to improve the manuscript.
|
|
|
|
COMMENT-1 |
The state-of-art is done in this review paper is far from enough.
|
RESPONSE: |
Respected Reviewer, Thank you for your suggestions and comments to improve the manuscript. The manuscript is modified according to the suggested changes.
|
|
|
COMMENT-2 |
A potential plagiarism issue is detected.
|
RESPONSE: |
Respected Reviewer, Thank you for your suggestions and comments to improve the manuscript. The manuscript is modified according to the suggested changes. All the figures are modified and redrawn as per your concerns and cited appropriately. The manuscript is not falling into any plagiarism. Thank you very much for giving valuable suggestions. · We authors are updated figures 4 to 9 according to the respected reviewer's suggestions and the same are represented below for your consideration. Respected reviewer, thanks for understanding our concerns.
Figure 5. Block diagram of the BMS[50] Figure 6. The general architecture of SOC system[50] Figure 7. Categorization of estimation SOC methods[50] Figure 8. OCV vs SOC tested under 25 °C[50] Figure 9. The comprehensive structure of Neural Network for SOC estimation [50] |
|
|
COMMENT-3 |
Similar work had been done intensively and NO new technical significance or point of view is provided in this paper.
|
RESPONSE: |
Respected Reviewer, Thank you for your valuable suggestion. The novelty of the proposed work is as follows; The article presents advanced SOC estimating techniques such as LSTM, GRU, CNN-LSMT, and hybrid techniques to estimate the average error of the SOC. A detailed comparison is presented with merits and demerits and it helps researchers in the implementation of EV applications. An effective Battery Management System (BMS) is essential to improve the battery performance, including charging-discharging control, precise monitoring, heat management, battery safety, protection, and also an accurate estimation of the State of Charge (SOC). The SOC is required to provide the driver with a precise indication of the remaining range. At present, different types of estimation algorithms are available, but they still have several challenges due to their performance degradation, complex electrochemical reactions, and inaccuracy. The estimating technique, average error, advantages, and disadvantages are examined methodically and independently in this paper. This research also identifies several factors, challenges, and potential recommendations for enhanced BMS and efficient estimating approaches for future sustainable EV applications. |
|
|
COMMENT-4 |
Overall, similar review work had been done extensively but this paper fails to provide a new perspective. Furthermore, most of the work done in this paper is copied from others, causing the potential plagiarism issue.
|
RESPONS0E: |
Respected Reviewer, Thank you for your suggestions and comments to improve the manuscript. The manuscript is modified according to the suggested changes. · The article presents advanced SOC estimating techniques such as LSTM, GRU, CNN-LSMT, and hybrid techniques to estimate the average error of the SOC. A detailed comparison is presented with merits and demerits and it helps researchers in the implementation of EV applications. · The manuscript is modified according to the suggested changes. All the figures are modified and redrawn as per your concerns and cited appropriately. The manuscript is not falling into any plagiarism. Thank you very much for giving valuable suggestions.
|

Reviewer 2 Report
Below are my constructive criticisms of the article.
- I recommend to modify title to "A Review on Different State of Battery Charge Estimation techniques and Management System for EV Applications".
- The abstract should be rewrite and define only aim of current as compared to previous review articles related to this field.
- Introduction does not need any table information, I suggest to introduce your work and article contents as section 1.
- Remaining all studies by author is acceptable in current form
Author Response

(The authors gave the same response as above.)

Reviewer 3 Report
The writing and organization of this work are good. Both Abstract and conclusions present a clear idea about this study. In this study, the authors discuss the analysis of Li-ion battery state of charge (SOC) and its management system. The importance of BMS is described in detail to achieve reliable and safe operating of Li-ion batteries. This paper reviews the conventional and advanced SOC estimation techniques and provides potential recommendations for developing advanced BMS and efficient estimation methods for future sustainable EV applications. It includes a complete explanation of each method's drawbacks, benefits, and estimation error. I don’t have any specific comments as the authors have presented this study very well. I have found a few issues which can improve the quality of this work.
- In the Abstract, what is BMS? Authors should define each abbreviation at the first place for better readability.
- Why did you select the time span from 2006 to 2022? Why did you select only the Web of Science database to select relevant studies?
- Authors consider the search criterion as "state of charge," followed by "Li-ion battery." Authors should check search key terms ideas from different systematic studies. I have concerns that the authors might have missed some studies by using this key term.
- Where are the Exclusion and Inclusion criteria for this work?
- In the Introduction section, authors should add a Table and compare their work with any relevant reported studies.
- Line 106-108, the proper reference is required for the given information.
- Line 122, what is SOH, SOE, and SOP? Please define at the first place of appearance to improve understanding for readers.
- Typo in line 196, it seems extra spacing is given.
- Section 3.5.2, I can notice a lack of discussion and reference literature on Hybrid techniques. The authors should add more sufficient details here. Authors must add a Table so readers can fastly understand the used Hybrid techniques and the research contribution of each reported study.
- Line 480, authors should recheck the sentence and remove possible writing typos.
- Section 5, it is suggested to provide sufficient discussion about potential challenges and recommendations or solutions separately. In this way, readers can easily focus on each challenge and possible solutions.
- Authors should check the Future Scope section as Figure 18 is not described in this section.
- The presentation quality of this work is nice. The research contribution of this study is justified.
- In the end, the Abbreviation Section must be revised as the authors have missed adding some abbreviations here.
Author Response
To
The Managing Editor,
Electronics,
MDPI.
Dear Editor,
Sub.: Submission of revision of paper electronics-1703990, A Review on Different State of Battery Charge Estimation techniques and Management System for EV Applications –Reg.
We would like to thank the Managing Editor, Handling Editor and reviewers for their valuable decision and recommendations made to improve the quality of the paper. We are happy to submit the revised manuscript entitled “A Review on Different State of Battery Charge Estimation techniques and Management System for EV Applications”, which we would like to submit for publication as a review article in Electronics. We thank you for providing such an opportunity, and we also thank the anonymous reviewers for their valuable and helpful comments. Based on the set of reviews the manuscript is thoroughly revised and the appropriate corrections have been made. The point-by-point response to the reviewers’ comments is included here forth for your kind perusal.
Thank you. We look forward to your positive response.
Yours sincerely,
Dr. Dhanamjayulu C
Assistant Professor (Senior)
School of Electrical Engineering,
Vellore Institute of Technology,
Vellore-632014. Tamilnadu, India.
E-mail: [email protected]
Author Responses:
We are very thank full to the editor and referee for the valuable time spent in reviewing this manuscript. We also thank them for their suggestions and comments to improve the manuscript. We have addressed all issues in the review report and believe that the revised version can meet the journal publication requirements.
Authors’ Responses to Reviewers’ Comments:
The authors, first of all, thank the reviewers for their valuable comments and suggestions.
RESPONSE TO REVIEWER’S COMMENTS - REVIEWER #3
The writing and organization of this work are good. Both Abstract and conclusions present a clear idea about this study. In this study, the authors discuss the analysis of Li-ion battery state of charge (SOC) and its management system. The importance of BMS is described in detail to achieve reliable and safe operating Li-ion batteries. This paper reviews the conventional and advanced SOC estimation techniques and provides potential recommendations for developing advanced BMS and efficient estimation methods for future sustainable EV applications. It includes a complete explanation of each method's drawbacks, benefits, and estimation error. I don’t have any specific comments as the authors have presented this study very well. I have found a few issues which can improve the quality of this work.
|
|||||||||||||||||||||||||||||||||||||||||||||||||||||||||||||||||||||||||
RESPONSE: Respected Reviewer, Thank you for your suggestions and comments to improve the manuscript. The manuscript is modified according to the suggested changes.
|
|||||||||||||||||||||||||||||||||||||||||||||||||||||||||||||||||||||||||
COMMENT-1 |
In the Abstract, what is BMS? Authors should define each abbreviation at the first place for better readability.
|
||||||||||||||||||||||||||||||||||||||||||||||||||||||||||||||||||||||||
RESPONSE: |
Respected Reviewer, Thank you for your suggestions and comments to improve the manuscript. The manuscript is modified according to the suggested changes.
We authors defined BMS as “Battery Management System” in the abstract, page 1, line 13, and also defined all abbreviations in the first place as per your suggestions. We authors would like to express sincere thanks to the reviewer for your valuable suggestion.
|
||||||||||||||||||||||||||||||||||||||||||||||||||||||||||||||||||||||||
|
|
||||||||||||||||||||||||||||||||||||||||||||||||||||||||||||||||||||||||
COMMENT-2 |
Why did you select the time span from 2006 to 2022? Why did you select only the Web of Science database to select relevant studies?
|
||||||||||||||||||||||||||||||||||||||||||||||||||||||||||||||||||||||||
RESPONSE: |
Respected Reviewer, Thank you for your suggestions and comments to improve the manuscript. · We authors considered all related SOC estimating techniques for the EV applications. We authors have taken into account related articles published in a web of science and other journals also. Respected reviewer, thanks for understanding our concern. |
||||||||||||||||||||||||||||||||||||||||||||||||||||||||||||||||||||||||
|
|
||||||||||||||||||||||||||||||||||||||||||||||||||||||||||||||||||||||||
COMMENT-3 |
Authors consider the search criterion as "state of charge," followed by "Li-ion battery." Authors should check search key terms ideas from different systematic studies. I have concerns that the authors might have missed some studies by using this key term.
|
||||||||||||||||||||||||||||||||||||||||||||||||||||||||||||||||||||||||
RESPONSE: |
Respected Reviewer, Thank you for your suggestions and comments to improve the manuscript. The manuscript is modified according to the suggested changes. The main objective of this work is to present the SOC estimation of Li-ion batteries for EV applications, it is one of the key factors for the implementation. We authors have focused only on SOC estimation techniques and presented comprehensively with merits and demerits. We authors have discussed some of the key futures in the future scope and it helps the researchers to develop EV. Respected reviewer, thanks for understanding our concerns.
|
||||||||||||||||||||||||||||||||||||||||||||||||||||||||||||||||||||||||
|
|
||||||||||||||||||||||||||||||||||||||||||||||||||||||||||||||||||||||||
COMMENT-4 |
Where are the Exclusion and Inclusion criteria for this work?
|
||||||||||||||||||||||||||||||||||||||||||||||||||||||||||||||||||||||||
RESPONSE: |
Respected Reviewer, Thank you for your suggestions and comments to improve the manuscript. The article presents the SOC estimation for EV applications with recently published articles and it focuses on SOC estimation techniques with merits and demerits. Electric vehicles (EVs) have acquired significant popularity in recent decades due to their performance and efficiency. EVs are already largely acknowledged as the most promising solutions to global environmental challenges and CO2 emissions. Li-ion batteries are most frequently employed in EVs due to their various benefits. An effective Battery Management System (BMS) is essential to improve the battery performance, including charging-discharging control, precise monitoring, heat management, battery safety, protection, and also an accurate estimation of the State of Charge (SOC). The SOC is required to provide the driver with a precise indication of the remaining range. At present, different types of estimation algorithms are available, but they still have several challenges due to their performance degradation, complex electrochemical reactions, and inaccuracy. The estimating technique, average error, advantages, and disadvantages are examined methodically and independently in this paper. The article presents advanced SOC estimating techniques such as LSTM, GRU, CNN-LSMT, and hybrid techniques to estimate the average error of the SOC. A detailed comparison is presented with merits and demerits and it helps researchers in the implementation of EV applications. This research also identifies several factors, challenges, and potential recommendations for enhanced BMS and efficient estimating approaches for future sustainable EV applications.
|
||||||||||||||||||||||||||||||||||||||||||||||||||||||||||||||||||||||||
COMMENT-5 |
In the Introduction section, authors should add a Table and compare their work with any relevant reported studies.
|
||||||||||||||||||||||||||||||||||||||||||||||||||||||||||||||||||||||||
RESPONSE: |
Respected Reviewer, Thank you for your suggestions and comments to improve the manuscript. One of the reviewers is requesting to remove a table in the introduction section. In this regard, we authors have kept a table in the introduction section. In this conflict, we authors request to the reviewer provide necessary suggestions for the same. Respected reviewer, thanks for understanding our concerns. Table 3. Summary of recently reported studies covered and present review article covered
|
||||||||||||||||||||||||||||||||||||||||||||||||||||||||||||||||||||||||
COMMENT-6 |
Line 106-108, the proper reference is required for the given information.
|
||||||||||||||||||||||||||||||||||||||||||||||||||||||||||||||||||||||||
RESPONSE: |
Respected Reviewer, Thank you for your suggestions and comments to improve the manuscript. We authors added relevant references on lines 106-108, and updated the manuscript on the line108-110, as per the suggestion and represented below for your consideration.
BMS goods are advanced by a couple of corporations like Australian EV power, British REAP Organization, American Edition Company, Beijing Significant Power Technology, and Harbin Guantuo Power Equipment Cooperate limited [30]. |
||||||||||||||||||||||||||||||||||||||||||||||||||||||||||||||||||||||||
COMMENT-7 |
Line 122, what is SOH, SOE, and SOP? Please define at the first place of appearance to improve understanding for readers.
|
||||||||||||||||||||||||||||||||||||||||||||||||||||||||||||||||||||||||
RESPONSE: |
Respected Reviewer, Thank you for your suggestions and comments to improve the manuscript.
We authors defined State of Health (SOH), State of Energy (SOE), and State of Power (SOP). In addition to that, the authors also defined all abbreviations in the first place as per your suggestions. We authors would like to express sincere thanks to the reviewer for your valuable suggestion.
|
||||||||||||||||||||||||||||||||||||||||||||||||||||||||||||||||||||||||
COMMENT-8 |
Typo in line 196, it seems extra spacing is given.
|
||||||||||||||||||||||||||||||||||||||||||||||||||||||||||||||||||||||||
RESPONSE: |
Respected Reviewer, Thank you for your suggestions and comments to improve the manuscript.
We authors rectified the typo error in line 196 and updated the manuscript in line 199. We authors would like to express sincere thanks to the reviewer for your valuable suggestion. |
||||||||||||||||||||||||||||||||||||||||||||||||||||||||||||||||||||||||
COMMENT-9 |
Section 3.5.2, I can notice a lack of discussion and reference literature on Hybrid techniques. The authors should add more sufficient details here. Authors must add a Table so readers can fastly understand the used Hybrid techniques and the research contribution of each reported study.
|
||||||||||||||||||||||||||||||||||||||||||||||||||||||||||||||||||||||||
RESPONSE: |
Respected Reviewer, Thank you for your suggestions and comments to improve the manuscript.
We authors added a summary of the literature on hybrid techniques in the table and updated the manuscript in table 9, and the same is represented below for your consideration In [138] the SOC of the lithium-ion cell is adaptively estimated using the Multiple model adaptive estimation (MMAE) technique using a modified Enhanced self-correcting (ESC) cell model. When compared to the EKF result, SOC estimation converges more quickly. In [139] this study designs an enhanced Kalman filter (KF)-based adaptive observer by approximating the electrochemical model. The estimator's predictions are compared against experimental data in simulations. The simulation outcomes are more precise and efficient than KF. accuracy of this method is improved with <2% of error. In [140] EKF paired with an Adaptive neuro-fuzzy inference system (ANFIS) reduces error and improves accuracy over EKF alone. Root Mean Square Error (RMSE) compares EKF with EKF-assisted ANFIS. In this way, the hybrid technology improves precision and accuracy while reducing expenses. In [141] For Li-ion batteries with uncertain noise circumstances, a new noise adaptive moving horizon estimating (NAMHE) approach is suggested. The simulation outcomes show that the suggested technique reduces the SOC estimate error compared to the classic Moving horizon estimating (MHE) method. The RMSE of the suggested technique and MHE are 0.7543% and 1.3026 %, respectively. In [142] Different OCV test methodologies impact the correlation of the OCV-SoC; an effective OCV-SoC relationship may increase SoC online convergence speed and accuracy. AEKF SoC estimate technique was more accurate and reliable than EKF during driving cycles. 0.5481 % mean error of the proposed system. Hybrid methods give accurate outcomes and cost-effective. Table 10. Analysis of hybrid SOC estimation Algorithms.
|
||||||||||||||||||||||||||||||||||||||||||||||||||||||||||||||||||||||||
COMMENT-10 |
Line 480, authors should recheck the sentence and remove possible writing typos.
|
||||||||||||||||||||||||||||||||||||||||||||||||||||||||||||||||||||||||
RESPONSE: |
Respected Reviewer, Thank you for your suggestions and comments to improve the manuscript.
We authors modified the sentence below and same updated the manuscript. We authors would like to express sincere thanks to the reviewer for your valuable suggestion.
The average accuracy error is ≤ ±1 to ≤ ±2.19%, which is very less, and the best method for SOC estimation is as shown in Figure 11. |
||||||||||||||||||||||||||||||||||||||||||||||||||||||||||||||||||||||||
COMMENT-11 |
Section 5, it is suggested to provide sufficient discussion about potential challenges and recommendations or solutions separately. In this way, readers can easily focus on each challenge and possible solutions.
|
||||||||||||||||||||||||||||||||||||||||||||||||||||||||||||||||||||||||
RESPONSE: |
Respected Reviewer, Thank you for your suggestions and comments to improve the manuscript.
We authors provided the information on Challenges, causes, and recommendations to monitor the SOC in section 5, table 13. We authors updated the manuscript and would like to express sincere thanks to the reviewer for your valuable suggestion.
Due to performance degradation and complex electrochemical reactions, developing and arranging a Li-ion battery organization system for EVs is a top priority. Moreover, most well-defined battery experiments are performed in a controlled laboratory environment with constant current, voltage, and temperature. Few analyses exist on battery performance in severe, hot, wet, and rainy conditions. External mass affects battery capacity. Unmodeled consequences add to unconsidered algorithms and models. Temperature, aging, cell unbalancing, hysteresis characteristics, battery model-ling, self-discharge, charge/discharge rate, etc. are also factors in battery performance decline. Temperature, aging, cell unbalancing, hysteresis characteristics, battery modeling, self-discharge, charge/discharge rate, etc. are also factors in battery performance decline. The work summarizes the key findings by applying aging modeling to four different Li-ion battery capacity loss datasets [138]. Figure 16 shows Li-ion battery cycle life versus temperature at different charge rates. Many researchers have proposed battery SOC models. Every model suffers from missing data for real-world EV applications. To accurately estimate battery states, complex calculations, high cost, and accuracy are issues. Table.10 lists SOC monitoring challenges, causes, and recommendations. Figure 17 explains battery anode aging. Table.11 Challenges, causes, and recommendations to monitor the SOC
|
||||||||||||||||||||||||||||||||||||||||||||||||||||||||||||||||||||||||
COMMENT-12 |
Authors should check the Future Scope section as Figure 18 is not described in this section.
|
||||||||||||||||||||||||||||||||||||||||||||||||||||||||||||||||||||||||
RESPONSE: |
Respected Reviewer, Thank you for your suggestions and comments to improve the manuscript. We authors thoroughly checked the future section and added the description of figure 18 and updated the manuscript according to the suggested changes. Thank you to the reviewer for your valuable suggestions.
Based on a comprehensive review of existing analyses on SOC estimation, this re-view makes several important recommendations for future research as shown in Figure 18. • Hybridizing algorithms: To achieve satisfactory SoC estimation performance, hybrid methods are highly recommended, in which multiple methods enhance each other. • Advanced sensing equipment: It is essential in developing high-precision sensors to improve current and voltage measurement accuracy for accurate SoC estimation. • Cloud computing technology: The real-time operation of intelligent algorithms and BMS controller schemes can be enhanced further with proper monitoring and analysis via the cloud storage and big data platform. • Embedded systems: Additional research is needed to create an embedded prototype with a low computational cost and small memory units. • High-performance processors: To accelerate the training operation, a GPU-based high-performance processor along with appropriate activation functions, excitable parameters, and training algorithms are necessary. • State monitoring for the battery packs: State estimation and fault diagnosis for battery packs must be evaluated to reduce cost, power loss, size, voltage stress, and improve equalization time and efficiency. • It is necessary to have a generalized validation and benchmark method for SOC estimation. |
||||||||||||||||||||||||||||||||||||||||||||||||||||||||||||||||||||||||
COMMENT-13 |
The presentation quality of this work is nice. The research contribution of this study is justified.
|
||||||||||||||||||||||||||||||||||||||||||||||||||||||||||||||||||||||||
RESPONSE: |
Respected Reviewer, We authors would like to express our sincere thanks to the respected reviewer for the suggestions and comments to improve the manuscript. |
||||||||||||||||||||||||||||||||||||||||||||||||||||||||||||||||||||||||
COMMENT-14 |
In the end, the Abbreviation Section must be revised as the authors have missed adding some abbreviations here.
|
||||||||||||||||||||||||||||||||||||||||||||||||||||||||||||||||||||||||
RESPONSE: |
Respected Reviewer, Thank you for your suggestions and comments to improve the manuscript.
We authors revised the manuscript thoroughly and added all the abbreviations at the end of the manuscript. Thank you to the reviewer for your valuable suggestions.
Abbreviations:
|

Reviewer 4 Report
The paper is generally well structured and presented. My main concerns are as follows:
1) The general writing needs improved. For an example, a lot of descriptions of the architecture and functionalities of BMS systems can be shortened or presented in a more concise fashion.
2) There are many formatting issue throughout the paper (e.g. very long paragraphs, incorrectly indented sentences after equations, format of keywords, extra dot after Table xx, uncapitalized word in the title, undefined abbreviation at its first appearance, broken sentences, uncapitalized table captions, etc.). I believe that it should be the authors' responsibility to thoroughly check and revise them.
3) There appears to be sentences directly taken from the cited literature. The authors should thoroughly check through the paper to avoid potential plagiarism issues.
4) Sources should be credited if information in the table was from literature (e.g. Tables 1, 2, 10)
5) page 2; under Table 1; there seems to be a missing word in "Lithium titanate (LTO)".
6) Table 2: what is the "unit" for Lifespan?
7) Figure 1: the authors need to state the scope of the literature survey. Does it include patents, conference papers, research reports, literature published in foreign languages? Also, year 2022 is not finished. So, it does not make sense to include 2022 in the figure. Otherwise, it can be misleading.
8) Table 3 needs to be improved as it is chaotic!
9) Equations (2) and (5): not all the symbols are defined.
10) Were Figures (3, 4, 5, 6, 7, 8, 9, 16, 17, 18) created by authors? If directly taken from literature, would you need permission to republish them?
Despite some criticism, I can understand how difficult it is to write a good review paper. Therefore, the authors' effort in presenting such a comprehensive review should also be commended.
Author Response
To The Managing Editor, Electronics, MDPI.
Dear Editor, Sub.: Submission of revision of paper electronics-1703990, A Review on Different State of Battery Charge Estimation techniques and Management System for EV Applications –Reg. We would like to thank the Managing Editor, Handling Editor and reviewers for their valuable decision and recommendations made to improve the quality of the paper. We are happy to submit the revised manuscript entitled “A Review on Different State of Battery Charge Estimation techniques and Management System for EV Applications”, which we would like to submit for publication as a review article in Electronics. We thank you for providing such an opportunity, and we also thank the anonymous reviewers for their valuable and helpful comments. Based on the set of reviews the manuscript is thoroughly revised and the appropriate corrections have been made. The point-by-point response to the reviewers’ comments is included here forth for your kind perusal.
Thank you. We look forward to your positive response.
Yours sincerely,
Dr. Dhanamjayulu C Assistant Professor (Senior) School of Electrical Engineering, Vellore Institute of Technology, Vellore-632014. Tamilnadu, India. E-mail: [email protected]
Author Responses: We are very thank full to the editor and referee for the valuable time spent in reviewing this manuscript. We also thank them for their suggestions and comments to improve the manuscript. We have addressed all issues in the review report and believe that the revised version can meet the journal publication requirements.
Authors’ Responses to Reviewers’ Comments: The authors, first of all, thank the reviewers for their valuable comments and suggestions.
RESPONSE TO REVIEWER’S COMMENTS - REVIEWER #4
|
|
COMMENT |
The paper is generally well structured and presented. My main concerns are as follows: |
RESPONSE: |
Respected Reviewer,
We authors would like to express our sincere thanks to the reviewer for finding interest in the manuscript. Thank you for your valuable recommendations to enhance the quality of the manuscript. The manuscript is modified according to the suggested changes. |
|
|
COMMENT-1 |
The general writing needs improved. For an example, a lot of descriptions of the architecture and functionalities of BMS systems can be shortened or presented in a more concise fashion.
|
RESPONSE: |
Respected Reviewer,
Thank you for your suggestions and comments to improve the manuscript. The manuscript is modified according to the suggested changes.
Currently, BMS is commonly employed by several vehicle companies, universities, and colleges. BMS goods are advanced by a couple of corporations like Australian EV power, British REAP Organization, American Edition Company, Beijing Significant Power Technology, and Harbin Guantuo Power Equipment Cooperate limited [30]. The application of BMS in EVs remains at the initial point. The base is that the quantity of batteries is 100 times above that of transportable devices in EVs [34-37]. Additionally, EVs are planned to supply high currents, voltage, and power. This process makes BMS extra tricky than portable electronics. The general role of a BMS is shown in Figure 3, it shows the general function of a BMS, and it consists of various kinds of actuators, sensors, signal lines, and controllers. Sample circuit measures temperature, voltage, and current affording to the gating sign achieved from the controller circuit. The vital work of the controller circuit is to estimate the SOC, State of Health (SOH), State of Energy (SOE), and State of Power (SOP) of batteries over progressive algorithms and analog signals. The battery measurements of voltage, temperature, and currents are changed. After that, the data will be communicated to the vehicular controller and supply significant choice issues for vehicular and power distribution [39-42]. The BMS analyses EV power distribution and energy storage faults. Many researchers have proposed battery models in various ways. In [43], Figure 4 shows the BMS section divided into software and hardware assembly. 2.1. BMS Hardware BMS uses various sensor frameworks to screen and measure battery parameters like current, temperature, and voltage. Some researchers propose EIS (Electrochemical Impedance Spectroscopy) to screen battery cell impedance [44]. High-cost devices and space limits make high-accuracy information outside the lab difficult to obtain. To stop overheating, charging, and discharging, a protection system must be developed. Constant voltage/current is used to charge batteries, and a galvanostat and potentiostat may be needed. Balance cells may also need a variable rheostat. Balancing cells is a key strategy for improving battery pack stability and estimating battery life. Temperature affects cell reliability, performance, and imbalance. Thus, [45] acknowledged that reducing temperature differences between cells is important and should be observed and worked on. BMS unit works independently after data/information transfer. A controlled transceiver is required to send data inside the BMS. With wireless telecommunication and smart batteries, charger and battery can share a wealth of information [46]. 2.2. BMS Software BMS software is the arrangement's midpoint. It controls sensor data and hardware operations to create choices and state approximations. BMS software must include sample rate, switch control checking in the cell balancing controller, sensor scheme, and uniform active security circuit strategy. Online processing and research are required to inform and regulate battery functions. Robust automated information analysis and reliability may be key because the study handles state assessment and fault finding. This information will be presented to the operator in an easy-to-use interface. Below are BMS-specific roles. Total cell voltage, current, separate cell voltage measurement, temperature, impedance, and smoke detection are battery parameters. Battery state estimation includes SOH and SOC, which group working situations supported by state-space representations, NN (Neural Networks), symbolic/fuzzy logic, etc. [47]. Cell balancing without over-discharging/charging maximizing battery performance. It aligns with SOC cell stages. The controller can control charging based on each cell's SOC. Thus, precise estimation of the SOC of each cell is required to improve cell balancing. Online processing will expose sensitive issues. Data analysis is needed to determine battery faults and out-of-tolerance conditions. Before potential problems, important information will be noted. The BMS interface must display vital data. On the control panel, the battery SOC shows the range. Also, irregular disturb-ing and extra ideas are wanted to inform the operators of the battery estimate and calculation [48-50]. Figure 5 shows the BMS block diagram. The working detail is broken down. The battery's measurement block converts current, temperature, and voltage into digital signals at each point. These constraints are used to evaluate the battery's SOH and SOC. A capability estimation block is used to control max charging/discharging current. The cell balance block uses the capability estimation results to limit over-discharge/charge irregularities. Ground fault-finding improves system safety. The thermal management lump monitors the temperature to ensure battery safety. An input- and the output-controlled transceiver is used. To receive and transmit massive amounts of data, a high-speed controlled transceiver is required. The present promoted various BMS separately play out the elemental capacities in an unpredicted mode. |
COMMENT-2 |
There are many formatting issue throughout the paper (e.g. very long paragraphs, incorrectly indented sentences after equations, format of keywords, extra dot after Table xx, uncapitalized word in the title, undefined abbreviation at its first appearance, broken sentences, uncapitalized table captions, etc.). I believe that it should be the authors' responsibility to thoroughly check and revise them.
|
RESPONSE: |
Respected Reviewer, Thank you for your suggestions and comments to improve the manuscript. The manuscript is modified according to the suggested changes. We authors thoroughly checked the manuscript and updated all the formatting issues, typo errors, incorrectly indented sentences after equations, the format of keywords, extra dot after Table, the uncapitalized word in the title, undefined abbreviation at its first appearance, broken sentences, and uncapitalized table captions in the manuscript according to the suggested changes. Thank you to the reviewer for your valuable suggestions. |
|
|
COMMENT-3 |
There appears to be sentences directly taken from the cited literature. The authors should thoroughly check through the paper to avoid potential plagiarism issues.
|
RESPONSE: |
Respected Reviewer, Thank you for your valuable suggestion.
We authors thoroughly checked the manuscript and modified the cited literature, updated the manuscript according to the suggested changes. Respected reviewer thanks for understanding our concern. |
|
|
COMMENT-4 |
Sources should be credited if information in the table was from literature (e.g. Tables 1, 2, 10)
|
RESPONS0E: |
Respected Reviewer, Thank you for your suggestions and comments to improve the manuscript. The manuscript is modified according to the suggested changes.
We authors have credited the information in tables 1, 2, and 10, which are taken from the literature by giving proper citations in the updated manuscript. Thank you to the reviewer for your valuable suggestions. |
|
|
COMMENT-5 |
page 2; under Table 1; there seems to be a missing word in "Lithium titanate (LTO)".
|
RESPONS0E: |
Respected Reviewer, Thank you for your suggestions and comments to improve the manuscript. The manuscript is modified according to the suggested changes. We authors corrected the “LTO” abbreviation and updated the manuscript to “Lithium titanium Oxide (LTO)”. We authors would like to express sincere thanks to the reviewer for your valuable suggestion. |
COMMENT-6 |
Table 2: what is the "unit" for Lifespan?
|
RESPONS0E: |
Respected Reviewer, Thank you for your suggestions and comments to improve the manuscript. The manuscript is modified according to the suggested changes. We authors updated the units of Lifespan is “hours” and same updated the manuscript. We authors would like to express sincere thanks to the reviewer for your valuable suggestion.
|
COMMENT-7 |
Figure 1: the authors need to state the scope of the literature survey. Does it include patents, conference papers, research reports, literature published in foreign languages? Also, year 2022 is not finished. So, it does not make sense to include 2022 in the figure. Otherwise, it can be misleading.
|
RESPONS0E: |
Respected Reviewer, Thank you for your suggestions and comments to improve the manuscript. The manuscript is modified according to the suggested changes. We authors modified figure 1 as per the suggested changes and represented it below for your consideration. Respected reviewer thanks for understanding our concern.
Figure 1. Number of research articles on Li-ion battery SOC estimation per year
|
COMMENT-8 |
Table 3 needs to be improved as it is chaotic!
|
RESPONS0E: |
Respected Reviewer, Thank you for your suggestions and comments to improve the manuscript. The manuscript is modified according to the suggested changes. We authors improved table 3 and updated the manuscript. Thank you to the reviewer for your valuable suggestions.
|
COMMENT-9 |
Equations (2) and (5): not all the symbols are defined.
|
RESPONS0E: |
Respected Reviewer, Thank you for your suggestions and comments to improve the manuscript. The manuscript is modified according to the suggested changes. We authors defined all the symbols as mentioned in equations 2 & 5 and updated the manuscript accordingly. Thank you to the reviewer for your valuable suggestions. (2) Where "ɳ" indicates coulombic efficiency, "i" indicates the current of the battery and "Cm " indicates total capacity. Measurement equation∶ ym= Cm xm+Dm um+zm (5) Where A, B, C & D represents the covariance matrixes, “x” is the system state, “f” represents the process noise, “u” represents the control input, “y” represents the measurement input, “z” represents measurement noise. |
COMMENT-10 |
Were Figures (3, 4, 5, 6, 7, 8, 9, 16, 17, 18) created by authors? If directly taken from literature, would you need permission to republish them?
|
RESPONS0E: |
Respected Reviewer, Thank you for your suggestions and comments to improve the manuscript. The manuscript is modified according to the suggested changes. All the figures are modified and redrawn as per your concerns and cited appropriately. The manuscript is not falling into any plagiarism. Thank you very much for giving valuable suggestions. · We authors are updated figures 3 to 18 according to the respected reviewer's suggestions and the same are represented below for your consideration. Respected reviewer, thanks for understanding our concerns. Figure 3. The general role of a BMS
Figure 4. The basic outline of BMS in EV Figure 5. Block diagram of the BMS Figure 6. The general architecture of the SOC system Figure 7. Categorization of estimation SOC methods Figure 8. OCV vs SOC tested under 25 °C Figure 9. The comprehensive structure of Neural Network for SOC estimation Figure 16. Temperature vs battery cycle life at a dissimilar charge rate of Li-ion battery [143]. Figure 17. Explanations for the aging of a battery at the anode [145]. Figure 18. Future trends in advanced BMS for EV applications
|
COMMENT-11 |
Despite some criticism, I can understand how difficult it is to write a good review paper. Therefore, the authors' effort in presenting such a comprehensive review should also be commended.
|
RESPONS0E: |
Respected Reviewer, Thank you for your suggestions and comments to improve the manuscript. The manuscript is modified according to the suggested changes.
|

Reviewer 5 Report
The article requires a profound reorganization of the contents, a scrupulous verification of the abbreviations (most are not defined at first use, and the abbreviations section is neither complete nor in alphabetical order), updating of the citations (125 are previous to 2018) and a thorough revision of English before it can be considered for publication.
Author Response
To The Managing Editor, Electronics, MDPI.
Dear Editor, Sub.: Submission of revision of paper electronics-1703990, A Review on Different State of Battery Charge Estimation techniques and Management System for EV Applications –Reg. We would like to thank the Managing Editor, Handling Editor and reviewers for their valuable decision and recommendations made to improve the quality of the paper. We are happy to submit the revised manuscript entitled “A Review on Different State of Battery Charge Estimation techniques and Management System for EV Applications”, which we would like to submit for publication as a review article in Electronics. We thank you for providing such an opportunity, and we also thank the anonymous reviewers for their valuable and helpful comments. Based on the set of reviews the manuscript is thoroughly revised and the appropriate corrections have been made. The point-by-point response to the reviewers’ comments is included here forth for your kind perusal.
Thank you. We look forward to your positive response.
Yours sincerely,
Dr. Dhanamjayulu C Assistant Professor (Senior) School of Electrical Engineering, Vellore Institute of Technology, Vellore-632014. Tamilnadu, India. E-mail: [email protected]
Author Responses: We are very thank full to the editor and referee for the valuable time spent in reviewing this manuscript. We also thank them for their suggestions and comments to improve the manuscript. We have addressed all issues in the review report and believe that the revised version can meet the journal publication requirements.
Authors’ Responses to Reviewers’ Comments: The authors, first of all, thank the reviewers for their valuable comments and suggestions.
RESPONSE TO REVIEWER’S COMMENTS - REVIEWER #5 |
|||||||||||||||||||||||||||||||||||||||||||||||||||||||||||||||||||||||||||||||||||||||||||||||||||
|
|||||||||||||||||||||||||||||||||||||||||||||||||||||||||||||||||||||||||||||||||||||||||||||||||||
COMMENT-1 |
The article requires a profound reorganization of the contents, a scrupulous verification of the abbreviations (most are not defined at first use, and the abbreviations section is neither complete nor in alphabetical order), and updating of the citations (125 are previous to 2018) and a thorough revision of English before it can be considered for publication.
|
||||||||||||||||||||||||||||||||||||||||||||||||||||||||||||||||||||||||||||||||||||||||||||||||||
RESPONSE: |
Respected Reviewer, Thank you for your suggestions and comments to improve the manuscript.
We authors revised the manuscript thoroughly and added all the abbreviations at the end of the manuscript. Thank you to the reviewer for your valuable suggestions. We checked grammar throughout the manuscript and also added recently published articles in the updated manuscript.
Abbreviations:
|
||||||||||||||||||||||||||||||||||||||||||||||||||||||||||||||||||||||||||||||||||||||||||||||||||
|
|

Round 2
Reviewer 1 Report
The reviewer would like to thank the authors' efforts in modifying the manuscript. However, the plagiarism issue which had been reported in the previous round has not yet been solved.
In detail, multiple figures are copied from the reference. Even though the authors claimed that the figures are redrawn, the authors fail to provide a different point of view from the references.
· Figure 5 is almost the same as the reference [50].
· Figure 6 is almost the same as the reference [50].
· Figure 7 is almost the same as the reference [50].
· Figure 8 is almost the same as the reference [50] and is very different from the reference [59] as claimed.
· Figure 9 is almost the same as the given reference [98].
Author Response
To
The Managing Editor,
Electronics,
MDPI.
Dear Editor,
Please find the invitation letter from MDPI electronics and cover letter can be find the below.
Dear Dr. Dhanamjayulu,
Thank you for your email.
I immediately applied to the editorial office after receiving your email, and got permission: we will offer a 300 CHF discount for this article.Sub.: Submission of revision of paper electronics-1703990, A Review on Different State of Battery Charge Estimation techniques and Management System for EV Applications –Reg.
We would like to thank the Managing Editor, Handling Editor, and reviewers for their valuable decision and recommendations made to improve the quality of the paper. We are happy to submit the revised manuscript entitled “A Review on Different State of Battery Charge Estimation techniques and Management System for EV Applications”, which we would like to submit for publication as a review article in Electronics. We thank you for providing such an opportunity, and we also thank the anonymous reviewers for their valuable and helpful comments. Based on the set of reviews the manuscript is thoroughly revised and the appropriate corrections have been made. The point-by-point response to the reviewers’ comments is included here forth for your kind perusal.
Thank you. We look forward to your positive response.
Yours sincerely,
Dr. Dhanamjayulu C
Assistant Professor (Senior)
School of Electrical Engineering,
Vellore Institute of Technology,
Vellore-632014. Tamilnadu, India.
E-mail: [email protected]
Author Responses:
We are very thank full to the editor and referee for the valuable time spent reviewing this manuscript. We also thank them for their suggestions and comments to improve the manuscript. We have addressed all issues in the review report and believe that the revised version can meet the journal publication requirements.
Authors’ Responses to Reviewers’ Comments:
The authors, first of all, thank the reviewers for their valuable comments and suggestions.
RESPONSE TO REVIEWER’S COMMENTS - REVIEWER #1
|
||||||||||||||||||||||||||||||||||||||||||||||||||||||||||||||||
COMMENT |
The reviewer would like to thank the authors' efforts in modifying the manuscript. However, the plagiarism issue which had been reported in the previous round has not yet been solved. In detail, multiple figures are copied from the reference. Even though the authors claimed that the figures are redrawn, the authors fail to provide a different point of view from the references.
|
|||||||||||||||||||||||||||||||||||||||||||||||||||||||||||||||
RESPONSE: |
Respected Reviewer, Thank you for your suggestions and comments to improve the manuscript
|
|||||||||||||||||||||||||||||||||||||||||||||||||||||||||||||||
COMMENT-1 |
· Figure 5 is almost the same as the reference [50]. · Figure 6 is almost the same as the reference [50]. · Figure 7 is almost the same as the reference [50]. · Figure 8 is almost the same as the reference [50] and is very different from the reference [59] as claimed. · Figure 9 is almost the same as the given reference [98].
|
|||||||||||||||||||||||||||||||||||||||||||||||||||||||||||||||
RESPONSE: |
Respected Reviewer, Thank you for your suggestions and comments to improve the manuscript. The manuscript is modified according to the suggested changes. All the figures are modified and redrawn as per your concerns. Thank you very much for giving valuable suggestions. • We authors are updated figures 5 to 9 according to the respected reviewer's suggestions and the same are represented below for your consideration. Respected reviewer, thanks for understanding our concerns.
Respected reviewer, We attached the Copyright form for Figure 9 at the bottom of this document for your reference.
Figure 5. Block diagram of the BMS.
Figure 6. The general architecture of the SOC system. Figure 7. Categorization of estimation SOC methods. Figure 8. OCV vs SOC tested under 25 °C.
Figure 9. The comprehensive structure of Neural Network for SOC estimation.
|
|||||||||||||||||||||||||||||||||||||||||||||||||||||||||||||||
RESPONSE TO REVIEWER’S COMMENTS - REVIEWER #3 |
||||||||||||||||||||||||||||||||||||||||||||||||||||||||||||||||
COMMENT |
Thank you for your detailed explanation. You have addressed all my concerns very well in this revised manuscript. The quality of this article has been significantly improved. In my view, the paper can be now accepted for publication. |
|||||||||||||||||||||||||||||||||||||||||||||||||||||||||||||||
RESPONSE: |
Respected Reviewer, We authors would like to express our sincere thanks to the respected reviewer for the suggestions and comments to improve the manuscript. Thank you very much.
|
|||||||||||||||||||||||||||||||||||||||||||||||||||||||||||||||
|
||||||||||||||||||||||||||||||||||||||||||||||||||||||||||||||||
COMMENT |
The paper has been well revised, which shows some improvements. Most of my concerns have been addressed by the authors. There are still some issues that need to be addressed (mostly minor). |
|||||||||||||||||||||||||||||||||||||||||||||||||||||||||||||||
RESPONSE: |
Respected Reviewer, Thank you for your suggestions and comments to improve the manuscript. |
|||||||||||||||||||||||||||||||||||||||||||||||||||||||||||||||
COMMENT-1 |
Tables 1-2: the unit of Voltage should be (V) not (v).
|
|||||||||||||||||||||||||||||||||||||||||||||||||||||||||||||||
RESPONSE: |
Respected Reviewer, Thank you for your suggestions and comments to improve the manuscript. The manuscript is modified according to the suggested changes. We authors updated the units of Voltage is “V” in table 1&2 and same updated the manuscript. We authors would like to express sincere thanks to the reviewer for your valuable suggestion.
Nominal Voltage (V)
|
|||||||||||||||||||||||||||||||||||||||||||||||||||||||||||||||
COMMENT-2 |
Table 3: it might be better to use "√" and "Χ" instead of "Yes" and "No". But I leave this to the authors' discretion. |
|||||||||||||||||||||||||||||||||||||||||||||||||||||||||||||||
RESPONSE: |
Respected Reviewer, Thank you for your suggestions and comments to improve the manuscript. The manuscript is modified according to the suggested changes. We authors updated table 3 as per your suggestions and the same is represented below for your consideration. We authors would like to express sincere thanks to the reviewer for your valuable suggestion.
Table 3. Summary of recently reported studies covered and present review article covered.
Note: √=Yes and x=No
|
|||||||||||||||||||||||||||||||||||||||||||||||||||||||||||||||
COMMENT-3 |
Figure 1: I still feel strongly that the "2022" should not be included as this may create an impression that the number of publications are dropping. This can be misleading. |
|||||||||||||||||||||||||||||||||||||||||||||||||||||||||||||||
RESPONSE: |
Respected Reviewer, Thank you for your suggestions and comments to improve the manuscript. The manuscript is modified according to the suggested changes. We authors modified figure 1 as per the suggested changes and represented it below for your consideration. Respected reviewer thanks for understanding our concern.
Figure 1. The number of research articles on Li-ion battery SOC estimation per year.
|
|||||||||||||||||||||||||||||||||||||||||||||||||||||||||||||||
|
|
|||||||||||||||||||||||||||||||||||||||||||||||||||||||||||||||
COMMENT-4 |
Figure 5: the quality of the figure is poor. |
|||||||||||||||||||||||||||||||||||||||||||||||||||||||||||||||
RESPONSE: |
Respected Reviewer, Thank you for your suggestions and comments to improve the manuscript. The manuscript is modified according to the suggested changes. Figure 5 is modified and redrawn as per your concerns and represented below for your consideration. Thank you very much for giving valuable suggestions.
Figure 5. Block diagram of the BMS.
|
|||||||||||||||||||||||||||||||||||||||||||||||||||||||||||||||
COMMENT-5 |
page 10: below equation (2): "Where" should rather be"where"
|
|||||||||||||||||||||||||||||||||||||||||||||||||||||||||||||||
RESPONSE: |
Respected Reviewer, Thank you for your suggestions and comments to improve the manuscript. The manuscript is modified according to the suggested changes. We authors updated the equation (2) and same updated the manuscript. We authors would like to express sincere thanks to the reviewer for your valuable suggestion.
|
|||||||||||||||||||||||||||||||||||||||||||||||||||||||||||||||
COMMENT-6 |
page 10: below equation (3): "Where" should be "where" and not indented. |
|||||||||||||||||||||||||||||||||||||||||||||||||||||||||||||||
RESPONSE: |
Respected Reviewer, Thank you for your suggestions and comments to improve the manuscript. The manuscript is modified according to the suggested changes. We authors updated the equation (3) and same updated the manuscript. We authors would like to express sincere thanks to the reviewer for your valuable suggestion.
|
|||||||||||||||||||||||||||||||||||||||||||||||||||||||||||||||
COMMENT-7 |
page 11: below equation (5): "Where" should be "where" |
|||||||||||||||||||||||||||||||||||||||||||||||||||||||||||||||
RESPONSE: |
Respected Reviewer, Thank you for your suggestions and comments to improve the manuscript. The manuscript is modified according to the suggested changes. We authors updated the equation (5) and same updated the manuscript. We authors would like to express sincere thanks to the reviewer for your valuable suggestion |
|||||||||||||||||||||||||||||||||||||||||||||||||||||||||||||||
COMMENT-8 |
Table 10: the table should be centered. |
|||||||||||||||||||||||||||||||||||||||||||||||||||||||||||||||
RESPONSE: |
Respected Reviewer, Thank you for your suggestions and comments to improve the manuscript. The manuscript is modified according to the suggested changes. We authors updated table 10 and same updated the manuscript. We authors would like to express sincere thanks to the reviewer for your valuable suggestion.
|
|||||||||||||||||||||||||||||||||||||||||||||||||||||||||||||||
COMMENT-9 |
Figure 16" caption: it should rather be " Battery cycle life versus temperature at a...". |
|||||||||||||||||||||||||||||||||||||||||||||||||||||||||||||||
RESPONSE: |
Respected Reviewer, Thank you for your suggestions and comments to improve the manuscript. The manuscript is modified according to the suggested changes. We authors updated the “Figure 16” caption as per your suggestions and the same is represented below for your consideration. We authors would like to express sincere thanks to the reviewer for your valuable suggestion.
Figure 16. Battery cycle life vs Temperature at a dissimilar charge rate of Li-ion battery.
|
|||||||||||||||||||||||||||||||||||||||||||||||||||||||||||||||
RESPONSE TO REVIEWER’S COMMENTS - REVIEWER #5 |
||||||||||||||||||||||||||||||||||||||||||||||||||||||||||||||||
COMMENT |
The authors addressed the reviewer's comments effectively. |
|||||||||||||||||||||||||||||||||||||||||||||||||||||||||||||||
RESPONSE: |
Respected Reviewer, We authors would like to express our sincere thanks to the respected reviewer for the suggestions and comments to improve the manuscript. Thank you very much. |

Reviewer 3 Report
Thank you for your detailed explanation. You have addressed all my concerns very well in this revised manuscript. The quality of this article has been significantly improved. In my view, the paper can be now accepted for publication.
Author Response
To
The Managing Editor,
Electronics,
MDPI.
Sub.: Submission of revision of paper electronics-1703990, A Review on Different State of Battery Charge Estimation techniques and Management System for EV Applications –Reg.
We would like to thank the Managing Editor, Handling Editor, and reviewers for their valuable decision and recommendations made to improve the quality of the paper. We are happy to submit the revised manuscript entitled “A Review on Different State of Battery Charge Estimation techniques and Management System for EV Applications”, which we would like to submit for publication as a review article in Electronics. We thank you for providing such an opportunity, and we also thank the anonymous reviewers for their valuable and helpful comments. Based on the set of reviews the manuscript is thoroughly revised and the appropriate corrections have been made. The point-by-point response to the reviewers’ comments is included here forth for your kind perusal.
Thank you. We look forward to your positive response.
Yours sincerely,
Dr. Dhanamjayulu C
Assistant Professor (Senior)
School of Electrical Engineering,
Vellore Institute of Technology,
Vellore-632014. Tamilnadu, India.
E-mail: [email protected]
Author Responses:
We are very thank full to the editor and referee for the valuable time spent reviewing this manuscript. We also thank them for their suggestions and comments to improve the manuscript. We have addressed all issues in the review report and believe that the revised version can meet the journal publication requirements.
Authors’ Responses to Reviewers’ Comments:
The authors, first of all, thank the reviewers for their valuable comments and suggestions.
RESPONSE TO REVIEWER’S COMMENTS - REVIEWER #3 |
|
COMMENT |
Thank you for your detailed explanation. You have addressed all my concerns very well in this revised manuscript. The quality of this article has been significantly improved. In my view, the paper can be now accepted for publication. |
RESPONSE: |
Respected Reviewer, We authors would like to express our sincere thanks to the respected reviewer for the suggestions and comments to improve the manuscript. Thank you very much.
|

Reviewer 4 Report
The paper has been well revised, which shows some improvements. Most of my concerns have been addressed by the authors. There are still some issues that need to be addressed (mostly minor).
1) Tables 1-2: the unit of Voltage should be (V) not (v)
2) Table 3: it might be better to use "√" and "Χ" instead of "Yes" and "No". But I leave this to the authors' discretion.
3) Figure 1: I still feel strongly that the "2022" should not be included as this may create an impression that the number of publications are dropping. This can be misleading.
4) Figure 5: the quality of the figure is poor.
5) page 10: below equation (2): "Where" should rather be "where"
6) page 10: below equation (3): "Where" should be "where" and not indented.
7) page 11: below equation (5): "Where" should be "where"
8) Table 10: the table should be centered.
9) Figure 16" caption: it should rather be "Battery cycle life versus temperature at a ...".
Author Response
To
The Managing Editor,
Electronics,
MDPI.
Sub.: Submission of revision of paper electronics-1703990, A Review on Different State of Battery Charge Estimation techniques and Management System for EV Applications –Reg.
We would like to thank the Managing Editor, Handling Editor, and reviewers for their valuable decision and recommendations made to improve the quality of the paper. We are happy to submit the revised manuscript entitled “A Review on Different State of Battery Charge Estimation techniques and Management System for EV Applications”, which we would like to submit for publication as a review article in Electronics. We thank you for providing such an opportunity, and we also thank the anonymous reviewers for their valuable and helpful comments. Based on the set of reviews the manuscript is thoroughly revised and the appropriate corrections have been made. The point-by-point response to the reviewers’ comments is included here forth for your kind perusal.
Thank you. We look forward to your positive response.
Yours sincerely,
Dr. Dhanamjayulu C
Assistant Professor (Senior)
School of Electrical Engineering,
Vellore Institute of Technology,
Vellore-632014. Tamilnadu, India.
E-mail: [email protected]
Author Responses:
We are very thank full to the editor and referee for the valuable time spent reviewing this manuscript. We also thank them for their suggestions and comments to improve the manuscript. We have addressed all issues in the review report and believe that the revised version can meet the journal publication requirements.
Authors’ Responses to Reviewers’ Comments:
The authors, first of all, thank the reviewers for their valuable comments and suggestions.
RESPONSE TO REVIEWER’S COMMENTS - REVIEWER #4 |
||||||||||||||||||||||||||||||||||||||||||||||||||||||||||||||||
COMMENT |
The paper has been well revised, which shows some improvements. Most of my concerns have been addressed by the authors. There are still some issues that need to be addressed (mostly minor). |
|||||||||||||||||||||||||||||||||||||||||||||||||||||||||||||||
RESPONSE: |
Respected Reviewer, Thank you for your suggestions and comments to improve the manuscript. |
|||||||||||||||||||||||||||||||||||||||||||||||||||||||||||||||
COMMENT-1 |
Tables 1-2: the unit of Voltage should be (V) not (v).
|
|||||||||||||||||||||||||||||||||||||||||||||||||||||||||||||||
RESPONSE: |
Respected Reviewer, Thank you for your suggestions and comments to improve the manuscript. The manuscript is modified according to the suggested changes. We authors updated the units of Voltage is “V” in table 1&2 and same updated the manuscript. We authors would like to express sincere thanks to the reviewer for your valuable suggestion.
NominalVoltage (V)
|
|||||||||||||||||||||||||||||||||||||||||||||||||||||||||||||||
COMMENT-2 |
Table 3: it might be better to use "√" and "Χ" instead of "Yes" and "No". But I leave this to the authors' discretion. |
|||||||||||||||||||||||||||||||||||||||||||||||||||||||||||||||
RESPONSE: |
Respected Reviewer, Thank you for your suggestions and comments to improve the manuscript. The manuscript is modified according to the suggested changes. We authors updated table 3 as per your suggestions and the same is represented below for your consideration. We authors would like to express sincere thanks to the reviewer for your valuable suggestion.
Table 3. Summary of recently reported studies covered and present review article covered.
Note: √=Yes and x=No
|
|||||||||||||||||||||||||||||||||||||||||||||||||||||||||||||||
COMMENT-3 |
Figure 1: I still feel strongly that the "2022" should not be included as this may create an impression that the number of publications is dropping. This can be misleading. |
|||||||||||||||||||||||||||||||||||||||||||||||||||||||||||||||
RESPONSE: |
Respected Reviewer, Thank you for your suggestions and comments to improve the manuscript. The manuscript is modified according to the suggested changes. We authors modified figure 1 as per the suggested changes and represented it below for your consideration. Respected reviewer thanks for understanding our concern.
Figure 1. The number of research articles on Li-ion battery SOC estimation per year.
|
|||||||||||||||||||||||||||||||||||||||||||||||||||||||||||||||
COMMENT-4 |
Figure 5: the quality of the figure is poor. |
|||||||||||||||||||||||||||||||||||||||||||||||||||||||||||||||
RESPONSE: |
Respected Reviewer, Thank you for your suggestions and comments to improve the manuscript. The manuscript is modified according to the suggested changes. Figure 5 is modified and redrawn as per your concerns and represented below for your consideration. Thank you very much for giving valuable suggestions.
Figure 5. Block diagram of the BMS.
|
|||||||||||||||||||||||||||||||||||||||||||||||||||||||||||||||
COMMENT-5 |
page 10: below equation (2): "Where" should rather be"where"
|
|||||||||||||||||||||||||||||||||||||||||||||||||||||||||||||||
RESPONSE: |
Respected Reviewer, Thank you for your suggestions and comments to improve the manuscript. The manuscript is modified according to the suggested changes. We authors updated the equation (2) and same updated the manuscript. We authors would like to express sincere thanks to the reviewer for your valuable suggestion.
|
|||||||||||||||||||||||||||||||||||||||||||||||||||||||||||||||
COMMENT-6 |
page 10: below equation (3): "Where" should be "where" and not indented. |
|||||||||||||||||||||||||||||||||||||||||||||||||||||||||||||||
RESPONSE: |
Respected Reviewer, Thank you for your suggestions and comments to improve the manuscript. The manuscript is modified according to the suggested changes. We authors updated the equation (3) and same updated the manuscript. We authors would like to express sincere thanks to the reviewer for your valuable suggestion.
|
|||||||||||||||||||||||||||||||||||||||||||||||||||||||||||||||
COMMENT-7 |
page 11: below equation (5): "Where" should be "where" |
|||||||||||||||||||||||||||||||||||||||||||||||||||||||||||||||
RESPONSE: |
Respected Reviewer, Thank you for your suggestions and comments to improve the manuscript. The manuscript is modified according to the suggested changes. We authors updated the equation (5) and same updated the manuscript. We authors would like to express sincere thanks to the reviewer for your valuable suggestion.
|
|||||||||||||||||||||||||||||||||||||||||||||||||||||||||||||||
COMMENT-8 |
Table 10: the table should be centered. |
|||||||||||||||||||||||||||||||||||||||||||||||||||||||||||||||
RESPONSE: |
Respected Reviewer, Thank you for your suggestions and comments to improve the manuscript. The manuscript is modified according to the suggested changes. We authors updated table 10 and same updated the manuscript. We authors would like to express sincere thanks to the reviewer for your valuable suggestion.
|
|||||||||||||||||||||||||||||||||||||||||||||||||||||||||||||||
COMMENT-9 |
Figure 16" caption: it should rather be " Battery cycle life versus temperature at a...". |
|||||||||||||||||||||||||||||||||||||||||||||||||||||||||||||||
RESPONSE: |
Respected Reviewer, Thank you for your suggestions and comments to improve the manuscript. The manuscript is modified according to the suggested changes. We authors updated the “Figure 16” caption as per your suggestions and the same is represented below for your consideration. We authors would like to express sincere thanks to the reviewer for your valuable suggestion.
Figure 16. Battery cycle life vs Temperature at a dissimilar charge rate of Li-ion battery.
|

Reviewer 5 Report
The authors addressed the reviewer's comments effectively.
Author Response
To
The Managing Editor,
Electronics,
MDPI.
Sub.: Submission of revision of paper electronics-1703990, A Review on Different State of Battery Charge Estimation techniques and Management System for EV Applications –Reg.
We would like to thank the Managing Editor, Handling Editor, and reviewers for their valuable decision and recommendations made to improve the quality of the paper. We are happy to submit the revised manuscript entitled “A Review on Different State of Battery Charge Estimation techniques and Management System for EV Applications”, which we would like to submit for publication as a review article in Electronics. We thank you for providing such an opportunity, and we also thank the anonymous reviewers for their valuable and helpful comments. Based on the set of reviews the manuscript is thoroughly revised and the appropriate corrections have been made. The point-by-point response to the reviewers’ comments is included here forth for your kind perusal.
Thank you. We look forward to your positive response.
Yours sincerely,
Dr. Dhanamjayulu C
Assistant Professor (Senior)
School of Electrical Engineering,
Vellore Institute of Technology,
Vellore-632014. Tamilnadu, India.
E-mail: [email protected]
Author Responses:
We are very thank full to the editor and referee for the valuable time spent reviewing this manuscript. We also thank them for their suggestions and comments to improve the manuscript. We have addressed all issues in the review report and believe that the revised version can meet the journal publication requirements.
Authors’ Responses to Reviewers’ Comments:
The authors, first of all, thank the reviewers for their valuable comments and suggestions.
RESPONSE TO REVIEWER’S COMMENTS - REVIEWER #5 |
|
COMMENT |
The authors addressed the reviewer's comments effectively. |
RESPONSE: |
Respected Reviewer, We authors would like to express our sincere thanks to the respected reviewer for the suggestions and comments to improve the manuscript. Thank you very much. |

Round 3
Reviewer 1 Report
The authors had well-addressed all the reviewer’s comments. I appreciate the effort the authors have put into their revisions.